# HERMESv3, a stand-alone multiscale atmospheric emission modelling framework - Part 1: global and regional module.

Marc Guevara[1], Carles Tena[1], Manuel Porquet[1], Oriol Jorba[1], Carlos Pérez García-Pando[1]

[1]Earth Sciences Department, Barcelona Supercomputing Center, Barcelona, 08034, Spain

*Correspondence to*: Marc Guevara (marc.guevara@bsc.es)

**Abstract.** We present the High-Elective Resolution Modelling Emission System version 3 (HERMESv3), an open source, parallel and stand-alone multiscale atmospheric emission modelling framework that computes gaseous and aerosol emissions for use in atmospheric chemistry models. HERMESv3 is coded in Python and consists of a *global_regional* module and a *bottom_up* module that can be either combined or executed separately. In this contribution (Part 1) we describe the

*global_regional* module, a customizable emission processing system that calculates emissions from different sources, regions and pollutants on a user-specified global or regional grid. The user can flexibly define combinations of existing up-to-date global and regional emission inventories and apply country specific scaling factors and masks. Each emission inventory is individually processed using user-defined vertical, temporal and speciation profiles that allow obtaining emission outputs compatible with multiple chemical mechanisms (e.g. Carbon-Bond 05). The selection and combination of emission inventories

and databases is done through detailed configuration files providing the user with a widely applicable framework for designing, choosing and adjusting the emission modelling experiment without modifying the HERMESv3 source code. The generated emission fields have been successfully tested in different atmospheric chemistry models (i.e. CMAQ, WRF-Chem and NMMB-MONARCH) at multiple spatial and temporal resolutions. In a companion article (Part 2) we describe the *bottom_up* module, which estimates emissions at the source level (e.g. road link) combining state-of-the-art bottom-up methods with local activity

and emission factors.

# 1    Introduction

Emission inputs of trace gases and aerosols play a key role in the performance of atmospheric chemistry models for air quality research and forecasting applications. Depending on the purpose of the application, an atmospheric chemistry model may be applied at global, regional or local (urban) scales. Similarly, the level of coverage and detail required for the emission input data will vary according to the type of study and modelling scale (e.g. Borge et al., 2014).

For global and regional modelling, emissions are typically estimated at country level (combining national statistics and technology-dependent emission factors), and then disaggregated using spatial proxies such as population density and land use. Different global and regional emission inventories are continuously being developed and made publicly available by research groups and international programs such as the Global Emissions Initiative (GEIA) (Frost et al., 2013). These inventories usually report total annuals per primary pollutant and source sector distributed over a rectangular grid at resolutions ranging from 1º by 1º to 0.1º by 0.1º. The practical use of these inventories suffers from several problems. On the one side, the reporting format is not directly compatible with the emission input requirements of atmospheric chemistry models as these typically ingest hourly and chemical species-based emissions over other grid projections and resolutions using specific file formats and conventions. On the other side, there are substantial discrepancies in the total emissions, sectorial emission shares, spatial distribution, and pollutant sources considered between the available inventories and therefore in their respective behaviour when used in atmospheric chemistry models (e.g. Granier et al., 2011; Trombetti et al., 2018; Saikawa et al., 2017). While having independent emission datasets instead of only one totally harmonized inventory is crucial from a science perspective, having the capacity to combine them and apply adjustment factors in a flexible and transparent way can be also of importance for air quality modelling studies. All in all, the incorporation of emission data into atmospheric chemistry models usually implies laborious programming in order to combine, adjust and adapt the original inventories to the model requirements.

Global and regional inventories are too imprecise for urban scale modelling applications (e.g. Timmermans et al., 2013). Emission and activity factors lack specificity for the local conditions of interest (e.g. Guevara et al., 2014), and the spatial proxies used to allocate the emissions are of poor resolution and may not apply to certain emission processes (e.g. Lopez-Aparicio et al., 2017). These inventories are for example limited when it comes to predict and assess the impact of emission reduction measures upon local air quality such as the change of speed limits (e.g. Baldasano et al., 2010) or the penetration of new vehicle technologies (e.g. Soret et al., 2014). Consequently, working at the urban scale requires dedicated local emission inventories combining activity data collected at a fine spatial scale (e.g. point source, road links, household) with bottom-up detailed emission algorithms that represent the different factors influencing the emission processes (e.g. vehicle speed, outdoor temperature).

In this paper and a companion paper (Guevara et al. in preparation), we describe the newly developed High-Elective Resolution Modelling Emission System version 3 (HERMESv3). HERMESv3 is a multiscale, open-source emission modelling framework that consists of two independent modules that can be either combined or executed separately: (i) the *global_regional* module and (ii) the *bottom_up* module. The *global_regional* module is a customizable emission processing system that calculates emissions from different sources, regions and pollutants on a user-specified global or regional model grid. The user can easily define combinations of existing global and regional emission inventories, which are individually processed using vertical, temporal and speciation profiles, and apply regional scaling factors and masks. The generated emission fields have been tested for different chemical mechanisms and atmospheric chemistry models, including CMAQ (Appel et al., 2017), WRF-Chem (Grell et al., 2005) and NMMB-MONARCH (Badia et al., 2017) models, and can be easily adapted to other models, grids or chemical mechanisms upon demand.

The *bottom_up* module is an emission model that can be used to estimate emissions at the source level (e.g. road link, industrial facility, crop type) and hourly level combining state-of-the-art estimation methods with local activity and emission factors along with meteorological data. This model covers the estimation of bottom-up emissions from point sources (e.g. power and manufacturing industries), road transport, residential combustion and agricultural activities (manure management, fertilizer application and crop operations), as well as the modelling of detailed emission scenarios for air quality planning studies. Besides the aforementioned atmospheric chemistry models, the emission outputs of this module are also adapted for their application with the R-LINE urban dispersion model (Snyder et al., 2013).

We conceive HERMESv3 as a flexible multiscale modelling framework that allows integrating and combining different emissions estimation approaches, so that the emission related outputs can be as detailed and specific as possible for the different domains (global, regional or local) involved in the corresponding application.

The development of HERMESv3 is based on the knowledge acquired from previous versions of HERMES for Spain (Baldasano et al, 2008; Guevara et al., 2013), Europe (Ferreira et al., 2013) and Mexico City (Guevara et al., 2017) that have been developed at the Earth Sciences Department of the Barcelona Supercomputing Center (BSC) during the last decade. Other existing emission software such as HEMCO (Keller et al., 2014) and PREP-CHEM-SRC (Freitas et al., 2011) have also been taken as a reference for the development of HERMESv3.

In this paper (Part 1) we provide a description of the *global_regional* module (herein referred to as HERMESv3_GR). The *bottom_up* module is described in the companion paper (Part2; Guevara et al., in preparation). The paper is organized as follows. Section 2 describes the processing system and its main functionalities together with some illustrative examples of the outputs that can be generated with this tool. Section 3 describes some of the current implementations of HERMESv3_GR for air quality modelling. Finally, Section 4 presents the main conclusions of this work.

## 2    Description of HERMESv3

### 2.1    Overview

Figure 1 shows a schematic representation of the structure of HERMESv3_GR along with the execution workflow. HERMESv3_GR first defines the destination grid and selects the emission inventories (see Sect. 2.2), and the vertical, temporal

and speciation profiles based on the specifications defined by the user in the general and emission inventory configuration files (see Sect. 2.3 and 2.4, respectively). During the initialization process, HERMESv3_GR automatically creates a set of auxiliary files that are subsequently used during the emission calculation process. These auxiliary files, including the output grid description, the time zones and the country mask, are specific to each new working domain and are stored by default after their creation so that they can be reused in subsequent executions. The emissions are calculated in four steps that are applied to each

pollutant sector and species of the selected original emission inventories. These four steps include: (i) the spatial regridding from source grid to destination grid (see Sect. 2.5.1), (ii) the mass distribution over model vertical layers (see Sect. 2.5.2), (iii) the temporal disaggregation (see Sect. 2.5.3) and (iv) the speciation mapping depending on the selected gas phase and aerosol chemical mechanisms (see Sect. 2.5.4). The emission calculation can combine inventories that cover different geographic domains and/or emission sectors. To prevent spatial overlapping between inventories a masking functionality is included

during the regridding phase. The user can define country-specific masks that restrict the applicability of the original inventory to a given region, and country-specific scaling factors. Once the emissions have been processed, HERMESv3_GR writes the output file following the requirements and conventions of the atmospheric chemistry model selected by the user in the general configuration file (see Sect. 2.5.5).

For each grid cell $x$ and vertical layer $l$ on the destination domain, and requested output species $e$, HERMESv3_GR computes the output hourly emissions following Eq. (1).

$$E\_out(x,l)_e = \sum_{i=1}^{I} \sum_{s=1}^{S} \sum_{\bar{x}=1}^{\bar{X}} \{E\_in(\bar{x}) * RF(\bar{x}) * VF(\bar{x},l) * TF * SF\}_{\bar{e},s,i} , \qquad (1)$$

Where $E\_in(\bar{x})_{\bar{e},s,i}$ is the input emission flux (kg m$^{-2}$ s$^{-1}$) of the species $\bar{e}$ and pollutant sector $s$ reported by inventory $i$ on the

source grid cell $\bar{x}$. $RF(\bar{x})_{\bar{e},s,i}$ is the remapping weight value from source grid cell $\bar{x}$ to the destination grid cell $x$ associated to species $\bar{e}$ and pollutant sector $s$ of inventory $i$. $VF(\bar{x},l)_{\bar{e},s,i}$ is the vertical weight factor for layer $l$ and source grid cell $\bar{x}$ assigned to species $\bar{e}$ and pollutant sector $s$ of inventory $i$ (0 to 1). $TF_{\bar{e},s,i}$ is the temporal weight factor $t$ assigned to species $\bar{e}$ and pollutant sector $s$ of inventory $i$. $SF_{\bar{e},s,i}$ is the speciation factor assigned to species $\bar{e}$ and pollutant sector $s$ of inventory $i$. The final $E\_out(x,l)_e$ is hourly emission for output species $e$ in destination grid cell $x$, layer $l$ and is the sum of: (i) all $\bar{X}$ source

grid cells $\bar{x}$ that contribute to destination grid cell $x$, (ii) all $S$ employed pollutant sources $s$ and (iii) all $I$ used emission inventories $i$. The units of the output emissions will vary according to the atmospheric chemistry model selected by the user. $RF(\bar{x})_{\bar{e},s,i}$ and $TF_{\bar{e},s,i}$ are computed following Eq. (2) and Eq. (3), respectively.

$$RF(\bar{x})_{\bar{e},s,i} = W(\bar{x})_i * \{MK(\bar{x}) * SC(\bar{x})\}_{\bar{e},s,i} \ , \tag{2}$$

$$TF_{\bar{e},s,i} = \{M(m) * D(d) * H(h)\}_{\bar{e},s,i} \ , \tag{3}$$

Where $W(\bar{x})_i$ is the regridding weight value that describes how the source grid cell $\bar{x}$ contributes to the destination grid cell $x$
(0 to 1). $MK(\bar{x})_{\bar{e},s,i}$ is the masking factor assigned to species $\bar{e}$ and pollutant sector $s$ of inventory $i$ on the source grid cell $\bar{x}$ (1 or 0). $SC(\bar{x})_{\bar{e},s,i}$ is the scaling factor assigned to species $\bar{e}$ and pollutant sector $s$ of inventory $i$ on the source grid cell $\bar{x}$. $M(m)_{\bar{e},s,i}$ is the monthly factor for month $m$ assigned to species $\bar{e}$ and pollutant sector $s$ of inventory $i$ (0 to 12). $D(d)_{\bar{e},s,i}$ is the daily factor for day $d$ assigned to species $\bar{e}$ and pollutant sector $s$ of inventory $i$ (0 to 28,29,30 or 31 depending on the total number of days for month $m$). $H(h)_{\bar{e},s,i}$ is the hourly factor for hour $h$ assigned to species $\bar{e}$ and pollutant sector $s$ of inventory $i$ (0 to 24).

## 2.2    Emission data library and preprocessing

Table 1 lists all the global and regional inventories currently considered in the HERMESv3_GR emission data library. On demand, new emission datasets can be added. At global scale, the inventories proposed for anthropogenic emissions include the Air Pollutants and Greenhouse Gases Emission Database for Global Atmospheric Research (EDGAR v4.3.2_AP, Cripa et al., 2018, EDGARv4.3.2_VOC, Huang et al., 2017), the Community Emissions Data System (CEDS, Hoesly et al. 2018) and the datasets derived from the Task Force Hemispheric Transport of Air Pollution community (HTAPv2.2, Janssens-Maenhout et al., 2015) and the Evaluating the Climate and Air Quality Impacts of Short-Lived Pollutants project (ECLIPSEv5.a, Klimont et al., 2017). Also at global scale, biomass burning emissions are provided by the Global Fire Assimilation System (GFASv1.2, Kaiser et al. 2012), whereas open burning of domestic waste and volcanic degassing emissions can be estimated using the inventories reported by Wiedinmyer et al. (2014) and Carn et al. (2017), respectively. Two European regional anthropogenic emission inventories are also considered, namely the TNO-MACC_III (Kuenen et al., 2014) and the EMEP (Mareckova et al., 2017). The emission data library compiles gaseous ($NO_x$, CO, NMVOC, $SO_x$, $NH_3$) and particulate (PM10, PM2.5, BC, OC) air pollutant emissions. Depending on the inventory, NMVOC emissions are reported as a single category (e.g. ECLIPSEv5.a), by individual species (e.g. GFASv1.2) or following the 25 species groups as proposed within the Global Emission Inventory Activity (GEIA) (Olivier et al., 1996) (e.g. EDGARv4.3.2_VOC). Most of the inventories are reported at the monthly level and include time series with multiple base years (past, present and future).

For each inventory, a specific pre-processing function has been developed to rewrite the original datasets on a common format. All the gridded emission inventory input files used by HERMESv3_GR: (i) are in the Network Common Data Form (NetCDF) format (http://www.unidata.ucar.edu/software/netcdf/), (ii) adhere to the Climate and Forecast (CF1.6) Metadata Conventions, (iii) include information of the cell centroids, boundary coordinates and cell areas of the working domain (needed for the

conservative remapping, see Sect. 2.5.1), (iv) report emissions in the same units (kg m$^{-2}$ s$^{-1}$), (v) follow a unique pollutant naming convention (e.g. "nox_no2" for NO$_x$ emissions expressed as NO$_2$ and "nox_no" for NO$_x$ emissions expressed as NO) and (vi) follow a unique file data storage convention (Sect. 2.4). Exceptionally, point source emission inventories (e.g. volcanic degassing emissions) are stored in CSV files that include information on the name of each source (e.g. name of the volcanoes),

geographic coordinates, altitude of injection of the emissions (in meters) and total amount of annual emissions (in kg s$^{-1}$). For this type of inventory, no pre-processing function is needed and the user can directly provide the data in the required format.

All the pre-processing functions used to transform the original inventories are included in the HERMESv3_GR code repository. It is important to note that the original gridded emission inventories are not stored inside the HERMESv3_GR

database and that users need to download them from the corresponding data provider's platform (e.g. EDGAR inventories are obtained from http://edgar.jrc.ec.europa.eu/). This decision is based on the fact that: i) some of the emission inventories that HERMESv3_GR can process cannot be passed on to third parties without the data provider's consent and ii) we believe it is good practice that users access the original files through the official source of information, so that the data providers can monitor the usage of their datasets. With the aim of helping the users, the HERMESv3_GR wiki contains a section that provides

information of each emission inventory, including the official downloading website/contact person (see Sect. 5). This information is also included in a README section inside each pre-processing function.

HERMESv3_GR only includes anthropogenic, biomass burning and volcano emission inventories. Natural emissions such as biogenic NMVOCs, mineral dust aerosols, Ocean DMS or lightning and soil NO, which have functional dependencies on

meteorological variables, are assumed to be calculated online during the execution of the corresponding atmospheric chemistry model (e.g. NMMB-MONARCH dust module; Pérez et al., 2011) or using specific emission models (e.g. MEGANv2.1; Guenther et al., 2012).

## 2.3    General configuration file

The general configuration options (e.g. start and end date, output file name, working domain description) can be passed to

HERMESv3_GR via a configuration file, arguments or a combination of both. The arguments passed by command line takes priority from the ones that appear in the configuration file.

The general configuration file is divided in four different sections (see example in Appendix 1):

- General: this section defines the main paths of the processing system (i.e. input, output, data), the name of the output

emission file and time step configuration parameters, including start and end dates, temporal resolution (i.e. monthly, daily, hourly) and number and frequency of time steps (e.g. 24 time steps every 3 hours).

- Domain selection: this section defines the working grid where emissions will be calculated (e.g. spatial extension, horizontal and vertical description). Currently, HERMESv3_GR can calculate emissions on grids with the following map

projections: regular lat-lon for global domains and rotated lat-lon and lambert conformal conic for regional domains. Other coordinate systems and combinations (e.g. regular lat-lon for regional domains) could be added upon request. In this section of the configuration file, the user also selects the format of the output emission file. Currently, HERMESv3_GR is able to write NetCDF emission output files following the CMAQ, WRF-Chem or NMMB-MONARCH conventions, and can be easily extended to other projections and atmospheric chemistry model conventions.

- Emission inventory configuration: this section defines the path to the file describing the configuration of the emission inventories (see Sect. 2.4).

- Profiles selection: this section defines the profile files that will be applied to perform the vertical distribution, temporal disaggregation and speciation treatment of the original emission inventories (see Sect. 2.5.1 to 2.5.4).

## 2.4 Emission inventory configuration file

The emission inventory configuration file allows the user to select the base emission inventories, pollutant sectors and species to combine and overlay for their simulations, and to choose the corresponding temporal, vertical and speciation profiles and optional scaling and masking factors that will be applied to the original emissions for their adaptation to the CTM requirements. Each line of the emission inventory configuration file belongs to a specific emission inventory, pollutant sector and pollutant species group, for which the user can define:

- Country-specific scaling factors that multiply the original emissions.
- Country-specific masks that restrict the applicability of the original inventory to a given region.
- A vertical profile to distribute the original emissions across the vertical layers of the working domain.
- A monthly, daily and hourly profile to temporally disaggregate the original emissions.
- A speciation profile to map the original pollutants species to a specific gas phase and aerosol chemical mechanism.

Figure 2 shows five examples of emission inventory configuration files and the resulting emission outputs calculated by HERMESv3_GR. The first column ("ei") indicates the name of the emission inventory, followed by the name of the pollutant sector ("sector"), the reference year of the emission inventory ("ref_year"), the requested pollutant species to be computed ("pollutants") and a field that indicates if this sector is activated or not ("active", 0 or 1). HERMESv3_GR combines all this information in order to select the corresponding file from the emission data library. In the first example (Fig. 2a), we selected the 2010 HTAPv2.2 organic carbon (OC) transport emissions, while in the second one (Fig. 2b) this inventory is combined with OC biomass burning emissions from GFASv1.2. The resulting output shows an increase of emissions in those areas typically affected by forest fires (e.g. Central Africa).

The following two columns of the configuration file are optional parameters that can be used to define country-specific scaling factors that multiply the original emissions ("factor_mask") and country-specific masks that restrict the applicability of the original emissions to the defined region ("regrid_mask"). Country-specific scaling factors are defined combining the ISO 3166-1 alpha-3 country code of the targeted country (https://unstats.un.org/unsd/tradekb/knowledgebase/country-code) with a

numerical factor. Scaling factors for more than one country need to be separated by a comma. Our third example (Fig. 2c) shows the original 2010 HTAPv2.2 OC transport emissions scaled by a factor of 5 in China and 0.5 in India (CHN 5, IND 0.5). On the other hand, country-specific masks are defined using the ISO 3166-1 alpha-3 country code preceded by either a "+" sign, which restricts the applicability of the inventory only to the targeted country, or a "-" sign, which restricts the applicability of the inventory to all the countries except the targeted one. The masks defined by the user can include more than

one country. In the fourth example (Fig. 2d), the HTAPv2.2 OC transport emissions are restricted to all countries except China and India (- CHN,IND), while in the fifth example (Fig. 2e) the OC transport emissions from ECLIPSEv5a are only applied to China and India (+ CHN,IND).

Column "frequency" defines the temporal resolution of the inventory (i.e. annual, monthly, daily). Column "path" defines the

root path of the emission files of each inventory. For all inventories, the root path consists of the common "<data_path>" defined in the general configuration file followed by the name of the institution providing the inventory, the name of the inventory and the temporal frequency. As shown in the first example, the root path of the HTAPv2.2 emission files is "<data_path>/jrc/htapv22/monthly_mean".

The alphanumeric codes specified in columns "p_vertical", "p_month" "p_day" "p_hour" and "p_speciation" refer to the vertical, monthly, daily, hourly and speciation profile IDs assigned to process the original emissions. All the codes are cross-referenced with text files where the vertical, temporal and speciation numerical factors are defined. As shown in the first example, the "p_hour" field allows the user to define specific diurnal profiles for weekdays, Saturdays and Sundays, which may be of relevance for certain pollutant sectors such as road transport (e.g. Mues et al., 2014). For the GFASv1.2 biomass

burning emissions (second example), the "p_vertical" field is not filled with a vertical profile ID but with two parameters that define: (i) the maximum altitude of the fire plume injection height ("method") and (ii) how the emissions are distributed across the layers below this maximum height ("approach") (see Sect. 2.5.2). Finally, the "comment" column is an optional field in which the user can add an observation.

## 2.5   Emission core module

The following sections describe the main functionalities of HERMESv3_GR, namely the spatial, vertical, temporal and speciation processing of the original emissions and the writing of the output file.

### 2.5.1 Spatial regridding

This function regrids the selected inventories from their original source grid to the user-defined destination grid. The regridding process consists of two steps. The first step uses the Earth System Modeling Framework (ESMF) regrid weight generation application (Hill et al., 2004) to calculate a regridding weight matrix that describes how points in the source grid contribute to points in the destination grid. The regridding method is first-order conservative, which means that it preserves the integral of the source field across the regridding. The weight for a particular source cell $i$ and destination cell $j$ ($W_{i,j}$) is based on the ratio of the source cell area overlapped with the corresponding destination cell area (Eq. 4):

$$W_{i,j} = f_{i,j} * \frac{AS_i}{AD_j} , \qquad (4)$$

Where $f_{i,j}$ is the fraction of the source cell $i$ contributing to destination cell $j$, and $AS_i$ and $AD_j$ are the areas of the source and destination cells.

The second step is the multiplication of the emissions on the source grid by the regridding weight matrix and, if previously defined by the user in the emission inventory configuration file, the corresponding scaling and/or masking factors to produce emissions on the destination grid. Country-specific scaling and masking factors are generated with a gridded country mask created during the initialization process. A current limitation of the masking method is that it does not consider that country border cells may include emissions of more than one country (i.e. it is assumed that all emissions belong to the country that contains the largest fraction of the cell). This limitation is mainly driven by the fact that most of the original inventories do not provide the information of the emitting country (i.e EDGAR, HTAP, ECLIPSE and CEDS report total emissions per grid cell but do not specify which fraction corresponds to which country). Future improvements will include the use of this information when given by the original inventory (i.e. EMEP and TNO_MACC-iii).

In the case of point source inventories (e.g. volcano degassing emissions) that are not reported on a regular grid but on specific lat-lon locations, the remapping is performed using a nearest destination to source approach. (When multiple source points are mapped into the same grid cell, the destination is the sum of the source emission values.) For point source emissions, neither scaling nor masking options are available, as the user can directly modify and/or erase individual point sources in the corresponding inventory input file.

The regridding process allows the user to remap the original emissions to global or regional grids with flexible spatial resolutions and several map projections, including regular lat-lon, rotated lat-lon, lambert conformal conic and mercator. Other map projections (e.g. polar stereographic) can potentially be added to the processing system in future releases. Figure 3 shows an example of the 0.1x0.1 degree HTAPv2.2 black carbon (BC) transport emissions regridded onto: (a) a 1x1.4 degree global

regular lat-lon domain, (b) a 0.1x0.1 degree regional rotated lat-lon domain, (c) a 50x50 km regional mercator grid and (d) a 4x4 km regional lambert conformal conic grid.

In its current version, HERMESv3_GR does not use any type of spatial proxy (e.g. land use, population data) during the remapping process. The main reason for this is that most of the inventories currently available in the emission data library have a spatial resolution that is higher and suitable enough for global and regional air quality modelling (i.e. 0.1x0.1 degrees). However, for those inventories with low spatial resolution (e.g. ECLIPSEv5a, 0.5x0.5 degree) the application of sector specific spatial proxies may be of importance when performing the remapping onto finer working domains. Future work will focus on improving this limitation by rebalancing the interpolation weights derived from ESMF with weight factors based on spatial proxies.

### 2.5.2 Vertical distribution

Once the emissions are allocated in the horizontal grid, the next step is to distribute them across the vertical layers of the destination domain. For this task, two input files are required: (i) a CSV file containing a description of the domain's vertical layers (i.e. approximate heights above the ground of the top of each vertical layer, in meters) and (ii) a CSV file containing a description of the vertical profile ID previously assigned by the user in the emission inventory configuration file (i.e. fraction of emissions assigned to each vertical layer, between 0 and 1). Using this information, HERMESv3_GR interpolates the original emissions to the modelling domain layers.

Note that HERMESv3_GR is currently designed as an off-line model and cannot use or take into account the variability of the vertical layer depth used by atmospheric chemistry models based on sigma vertical coordinates. Consequently, the system cannot distribute the emissions to the exact sigma levels of the models (which slightly vary in time and space) but to a set of fixed vertical levels that are close to them. This assumption is in line with previous modelling works (e.g. Mailler et al., 2013). Moreover, the impact of this limitation can be assumed to be minor when compared to the large uncertainty and variability associated with the emission vertical profiles available in the literature (e.g. Bieser et al., 2011).

Figure 4 shows a graphical example of how the vertical distribution is performed. In the example, the destination modelling domain is defined as 6 layers with top heights of 75, 140, 190, 500 and 1200 meters above ground level (m.a.g.l.). On the other hand, the proposed vertical profile ID (V001) indicates that 0% of the total emissions should be assigned between 0 and 100 m a.gl., 10% between 100 and 200 m.a.g.l and the remaining 90% between 200 and 1000 m a.g.l. Note that the number and description of the vertical layers used to define the vertical profiles do not have to match with the ones of the destination domain. HERMESv3_GR internally interpolates homogenously the original weight fractions to the modelling domain's layers taking into account the thickness of each layer.

The user is able to define and assign any vertical profile to any emission inventory/pollutant sector/pollutant species. Some suggested vertical profiles for the energy and manufacturing industry (Bieser et al., 2011) and the air traffic sectors (Olsen et al., 2013) are included in the HERMESv3_GR database.

For the GFASv1.2 biomass burning inventory, the vertical emission distribution is not performed with a fixed vertical profile but using two parameters that define: (i) the maximum altitude of the fire plume injection height ("method") and (ii) how the emissions are distributed across the layers below this maximum height ("approach"). The fire plume injection height is directly provided by GFASv1.2 following two different methods. The first method ("sofiev") is based on a semi-empirical parameterisation detailed in Sofiev et al. (2013). The second method ("prm") consist on a plume rise model described by

Paugam et al. (2015). Regarding the approach, users can also choose between two options. The first one ("uniform"), consist on distributing uniformly all the emissions across the layers below the maximum injection height. The second one ("50_top") indicates that 50% of all emissions are allocated in the vertical layer that intersects with the maximum injection height, and the other 50% are distributed uniformly across the layers below the maximum injection height. The two approaches are derived from the work by Veira et al., (2015), in which they perform a sensitivity analysis to see the impact of the vertical distribution

of forest fire emissions on black carbon concentrations. Although uniform vertical distributions are used in most modelling studies, some works have also showed that fires with high injection heights might emit a large fraction of the emissions into the upper part of the plumes (e.g. Luderer et al., 2006).

Similarly, in the case of point source emission inventories (e.g. volcano degassing), the vertical distribution is not defined

using a fixed vertical profile but with the injection height field included in the input inventory file, which can be adjusted individually for each point source. Emissions are distributed homogenously across all the layers below the defined injection height.

### 2.5.3 Temporal distribution

This process distributes temporally the emissions from their original resolution (e.g. annual) to the one defined by the user

(monthly, daily or hourly). The emissions are multiplied by the user-defined monthly, weekly and hourly weight factors, which are specified on separated CSV files with the corresponding profile ID (i.e. "MXXX", "DXXX" and "HXXX" for monthly, weekly and hourly profiles, "XXX" being a three-digit numeric code that starts at "001"). Alternatively, users can also provide the temporal profiles using gridded files, which contain specific weight factors for each grid cell.

As in the case of the vertical profiles, the user is left free to define and assign any temporal profile to each pollutant sector and species. The HERMESv3_GR database includes by default the monthly, daily and hourly temporal profiles reported by LOTOS-EUROS (Denier van der Gon et al., 2011), which are partially based on the GENEMIS project (Friedrich and Reiss, 2004) and Hodzic et al (2012).

HERMESv3_GR estimates emissions in Universal Time Coordinate (UTC). However, all the user-defined hourly temporal profiles need to be introduced in Local Standard Time (LST). For each cell of the destination grid and time step, HERMESv3_GR converts the UTC simulation date to the corresponding LST and assigns to it the adequate local temporal factor. This conversion is done using as a basis a time zone grid created during the initialization process. Having the time zone information of each cell allows HERMESv3_GR to take into account Daylight Saving Time (DST) changes, which do not necessarily occur on the same date every year and in every country.

Figure 5 shows an example of the 6-hourly evolution (00, 06, 12 and 18h UTC) of the ECLIPSEv5a $NO_x$ transport emissions for a 24h simulation performed on a 0.5x0.7 degree global grid for the 23rd of February 2015. It is observed how the diurnal variation of emissions in different cities is in line with their local time. For instance, at 00:00h UTC time (first time step of the simulation), emissions in China are at their morning peak (08:00h LST), whereas in Barcelona are at their minimum (01:00h LST) and in New York close to their afternoon peak time (19:00h LST).

The application of gridded profiles can be of importance for those emission sectors whose temporal variation is not uniform across the space due to local influences such as climatology conditions (e.g. the effect of temperature on residential combustion emissions) or sociodemographic patterns (e.g. the effect of farming practices on agricultural emissions). Figure 6 compares the monthly agricultural soil $NH_3$ emissions (March and June 2010) reported by EDGARv4.3.2 in East Asia when using its default temporal profile (Fig. 6.a and c) and when combined with updated gridded temporal weights that consider the effect of meteorology and crop calendars (Fig. 6.b and 6.d). These gridded profiles were derived from the monthly inventories reported by Zhang et al. (2018) for China and Paulot et al. (2014) for rest of the world, which seasonality is based on the temporal parametrizations reported by Skjøth et al. (2011).

Results show large differences between the two results, especially in China and India, the main emitter countries for this sector. According to Fig. 6.e, in China the default profile allocates most of the emissions in March, whereas the updated temporal profile gives more weight to the months of June and July. Similarly, the default profile presents a flat distribution over India, whereas the improved profile indicates a peak during the months of May and June (Fig. 6f). In both cases, the updated monthly distribution is more in line with the seasonality of the $NH_3$ volume mixing ratio derived from the NASA's Atmospheric Infrared Sounder (AIRS) instrument (Warner et al., 2017). The possibility offered by HERMESv3_GR to use gridded temporal profiles derived from meteorological parametrizations can be extended to other sources such as the residential combustion sector, for which the application of the heating degree day approach has been proved to be effective (e.g. Mues et al., 2014).

### 2.5.4    Speciation mapping

This process converts the pollutants provided in the original emission inventories to the species needed by the atmospheric chemistry model of interest and its corresponding gas phase and aerosol chemical mechanism. The conversion is performed using a speciation CSV file, in which the user defines mapping expressions between the source inventory pollutants and destination chemical species. Each mapping expression defines the pollutant-to-species relationships and factors for converting the input emissions pollutant to the desired model species.

These conversion factors are mass-based (i.e. g of chemical specie · g of source pollutant[-1]) for all source inventory pollutants except for NMVOC, which requires a specific approach (see paragraph below). The factors proposed for $NO_x$ assume a split of 0.9 for NO and 0.1 for $NO_2$ for all sectors (Houyoux et al., 2000) except for road transport and biomass burning, for which specific factors are derived from the works by Burling et al. (2010) and Rappenglueck et al. (2013). In the case of $PM_{2.5}$, the factors are derived from multiple sources of information including the particular matter SPECIATE (Simon et al., 2010) and SPECIEUROPE (Pernigotti et al., 2016) databases and the works by Visschedijk et al. (2007) and Reff et al. (2009). Source specific organic matter (OM) to OC fractions are derived from Klimont et al. (2017). For pollutants that have only one way of being speciated (e.g., mapping the CO pollutant to the CO species) a default factor of 1 is proposed for all sources and inventories. During the chemical speciation process, HERMESv3_GR also performs a conversion from mass to moles for the gas-phase species using a molecular weight CSV file included in the input database of the system. Note that for $NO_x$ two molecular weights are proposed since some inventories report emissions as NO ("nox_no", 30 g·mol[-1]) and some others as $NO_2$ ("nox_no2", 46 g·mol[-1]).

For NMVOC emissions reported as individual chemical compounds (e.g. $C_2H_4O$ in GFASv1.2) or following the GEIA 25 NMVOC groups (e.g. voc15 in EDGARv4.3.2_VOC), the proposed conversion factors are mole-based (i.e. mol of chemical specie · mol of source pollutant[-1]) and are derived from the mechanism-dependent mapping tables developed by Carter (2015). In this case, the conversion from mass to moles of original emissions is performed beforehand, and also using the information of the molecular weight CSV file.

Finally, for NMVOC emissions reported as a single category (i.e. as a sum of $n$ individual chemical compounds) (e.g. EMEP), the conversion factors proposed in HERMESv3_GR for each inventory $i$, pollutant sector $s$ and chemical species $\bar{e}$ ($SF_{\bar{e},s,i}$) were estimated as follows (Eq. 5):

$$SF_{\bar{e},s,i} = \sum_{j=1}^{n} \frac{X_{j,s}}{MW_j} * C_{j,\bar{e}} , \tag{5}$$

Where $X_{j,s}$ is the mass fraction of chemical compound $j$ to total NMVOC emissions for source $s$, $MW_j$ is the molecular weight of chemical compound $j$ and $C_{j,i}$ is the mole-based conversion factor of chemical compound $j$ to destination chemical species $\bar{e}$. $X_{j,s}$ values are obtained from the NMVOC SPECIATE database, while $MW_j$ and $C_{j,i}$ where obtained from Carter et al. (2015). The units of resulting proposed conversion factors is mol of chemical specie · g of source pollutant$^{-1}$.

Each line of the speciation CSV file corresponds to a specific profile, which is cross-referenced with the profile ID previously defined in the emission inventory configuration file (i.e "EXXX", "XXX" being a three-digit numeric code that starts at "001"). The columns of the file refer to the names of the destinations species, which need to match the atmospheric chemistry model registry names of the emission variables. The HERMESv3_GR database currently includes speciation profiles for the Carbon

Bond 05 (CB05, CB05e51) (Whitten et al., 2010) and the Regional Acid Deposition Model 2nd generation (RADM2) (Stockwell et al., 1990) gas-phase mechanisms, as well as the fifth and sixth-generation aerosol modules (AERO5, AERO6) (Roselle et al., 2008; Appel et al., 2017) and the Modal Aerosol Dynamics Model for Europe with the secondary organic aerosol model (MADE/SORGAM) aerosol mechanisms (Ackermann et al., 1998; Schell et al., 2001 As in the case of the temporal and vertical weight factors, the user can create its own speciation profiles using other sources of information.

As illustration, Table 2 shows two examples of proposed speciation profiles and corresponding mapping expressions included in the HERMESv3_GR database. The first one maps the original GFASv1.2 emission species to the CB05 gas-phase and AERO5 aerosol chemical mechanisms. As shown, original NO$_x$ (which are expressed as NO) are mapped to the CB05 species nitrogen monoxide (NO), nitrogen dioxide (NO$_2$) and nitrous acid (HONO) using mass-based conversion factors of 0.72

("nox_no*0.72"), 0.18 ("nox_no*0.18") and 0.1 ("nox_no*0.1") (Burling et al., 2010). The terminal olefin bond (OLE) CB05 species is composed of the following GFASv1.2 NMVOCs: $C_8H_{16}$, $C_5H_{10}$, $C_3H_6$, $C_4H_8$, $C_6H_{12}$ and 50% of other high alkanes ("c8h16+c5h10+c3h6+c4h8+c6h12+0.5*hialkanes"). On the other hand, the difference between total primary PM2.5 and carbonaceous species (OC and BC) is mapped to the other fine aerosols (PMFINE) AERO5 species ("pm2.5-oc-bc"). In the second example, the CEDS road transport emissions are mapped to the RADM2 gas-phase mechanism and the

MADE/SORGAM aerosol module. NO$_x$ emissions (which are originally reported as NO$_2$) are mapped to NO and NO2 using mass-based conversion factors of 0.84 ("nox_no2*0.84") and 0.16 ("nox_no2*0.16") (Rappenglueck et al., 2013). . The toluene (TOL) RADM2 species is estimated to be the sum of the voc14 (toluene) and 29.3% of the voc13 (benzene) GEIA groups ("0.293*voc13+voc14") (Carter, 2015). Total BC emissions are assumed to be 20% in nucleation mode (ECI, "bc*0.2") and 80% in accumulation mode (ECJ, "bc*0.8") (Tuccella et al., 2012). As shown in these examples, the mapping expressions

can combine different types of mathematical expressions (i.e. addition, subtraction, multiplication).

### 2.5.5 Writing module

The calculated emissions are written in NetCDF4 uncompressed files following the conventions of the selected atmospheric chemistry model. During this process, the following actions take place: (i) conversion of units, and (ii) inclusion of mandatory global attributes.

## 2.6 Technical implementation

HERMESv3_GR is coded using Python 2.7.X and requires numpy (>= 1.9.1), netCDF4 (>= 1.3.1), cdo (>= 1.3.3), pandas (>= 0.22.0), geopandas (>= 0.4.0), pyproj (>= 1.9.5.1), configargparse (>= 0.11.0), cf_units (>= 1.1.3), ESMPy (>= 7.1.0), holidays (>= 0.4.1), pytz (>= 2017.2), timezonefinder (>= 2.1.0), mpi4py (>= 3.0.0) and pytest (>= 3.6.1) Python libraries.

The emission core module of HERMESv3_GR is parallelized using a domain decomposition strategy. This approach is considered to be the most effective since emissions are computed independently for each destination grid cell and no communication between cells is needed during the calculation process (see Eq (1)). Moreover, applying domain decomposition also allows decreasing the memory consumption per computational node.

Figure 7 shows a schematic representation of the domain decomposition strategy applied in HERMESv3_GR. During the spatial regridding, the destination working domain is divided into vertical sections, maintaining each column undividable. The number of divisions is equal to the number of processors to be used (P_0, P_1, …), which is defined by the user. The emission regridding process is performed independently in each processor and for each vertical section. The maximum number of cores to be used is equal to half of the number of columns of the destination domain. This limitation is defined by the ESMF software, which needs, at least, two complete columns to perform the spatial regridding. The 2D regridded emissions are kept in memory until the writing operation. During this task, the vertical (v0, v1, …) and temporal (t0, t1, …) weight factors previously estimated in the vertical and temporal distribution functions are applied to each emission subdomain in order to transform the 2D arrays (longitude, latitude) into 4D arrays (time, vertical layer, longitude, latitude). This strategy allows reducing the time during which the memory consumption is higher. Finally, each worker process writes simultaneously its result to a common NetCDF4 file, which ensures the gathering of the different subsets of the working domain into a single output. Alternatively, the user can select the option of executing the writing function in serial mode (i.e. using only one processor).

A scalability test was performed using the supercomputer MareNostrum4, which is host by the BSC, in order to determine the capability of HERMESv3_GR to scale up the emission calculation process. MareNostrum 4 is a supercomputer based on Intel Xeon Platinum processors at 2.1 GHz from the Skylake generation. It is a Lenovo system composed of SD530 Compute Racks, an Intel Omni-Path high performance network interconnect and running SuSE Linux Enterprise Server as operating system. It consists of 48 racks housing 3456 nodes, each one equipped with 48 cores and 96Gb of memory (2Gb per core)

(www.bsc.es/marenostrum/marenostrum). HERMESv3_GR was executed using a number of cores from 1 to 510, doubling the number in each successive test until using all cores of a node (i.e. 1, 2, 4, 8, …, 48) and then adding 48 (a whole node) until 510 (i.e. 96, 144, …, 510). Two separate sets of tests were performed, one using the parallel writing function and another using the serial approach.

All the tests were performed using a rotated lat-lon destination grid of 0.1x0.1 degrees with 701 rows, 1021 columns and 48 vertical layers covering North Africa, Europe and the Middle East (Fig. 3b). Hourly CB05 and AERO5 speciated emissions were estimated for 24 time steps using as input all the available emission pollutants and sectors of the TNO_MACC_III (Europe) and HTAPv2.2 (rest of countries) inventories.

As shown in the stacked area chart of Fig. 7, the increased number of cores used in the simulations speeds up the computations. The total execution time decreases from 4,842.6s (1 core) to 1,247s (510 cores), the lowest value being observed when using 32 cores (800.7s). The most time demanding function changes according to the number of cores used. For 1 to 8 cores, most of the computational work is done during the spatial regridding (between 54% and 34%) and the temporal distribution (between

15   39% and 25%), whereas for the other cases (16 to 510 cores), the writing process increasingly becomes the main time consumer (up to 84% of the total time when using 510 cores). These results clearly indicate that the parallel writing function does not scale properly. The reason behind this behaviour comes from the fact that the netCDF4 Python library writes the results in row-major order (C-style), while during the spatial regridding ESMF divides the domain in vertical sections (column-major order, FORTRAN-style). For each vertical division, netCDF4 Python has to call the writing function as many times as the

20   number of rows that conform the domain. Subsequently, an increase of cores (i.e. an increase of vertical divisions) directly increases the execution time of the parallel writing process. The performance of the system when applying the serial writing approach (black line with markers) varies as a function of the processors used. For a low number of cores (i.e. 1 to 48), the parallel writing is faster than the serial one. Nevertheless, when using 96 processors or more, the serial writing becomes faster since its execution time remains almost constant, in contrast to what is experienced with the parallel approach. This fact allows

25   reducing the total execution time by a factor of up to 1.5 when using 510 cores. The potential disadvantage of using the serial writing is that for large emission experiments (i.e. large domains) the user may run into memory problems since all the data needs to be treated by a single processor. In the present test, we solved this issue by using all the memory resources of a compute node without sharing them with other users (i.e. 96Gb). Considering the advantages and disadvantages of each method, both the serial and parallel writing approaches are enabled in HERMESv3_GR.

The low performance of the parallel writing function will be addressed in future versions of HERMESv3_GR. For this, two strategies will be tested, including: (i) the integration of an I/O server that allows writing completed rows in row-major order and (ii) the use of other libraries specific for parallel writing (e.g. pnetcdf). Despite this shortcoming, the current parallelization

strategy allows HERMESv3_GR execution time to be minimized to less than 15 minutes per run (32 cores), which can be considered acceptable in an operational environment.

## 3    Implementations

HERMESv3_GR has been successfully tested in different atmospheric chemistry models. The system is currently implemented within the NMMB-MONARCH, which contributes to the multi-model ensemble forecasts of the International Cooperative for Aerosol Prediction (ICAP) ([www.nrlmry.navy.mil/aerosol/icap.1135.php](www.nrlmry.navy.mil/aerosol/icap.1135.php)). HERMESv3_GR has also been coupled with the CMAQ in the framework of the AIRE-CDMX air quality forecasting system for Mexico City (<http://www.aire.cdmx.gob.mx/pronostico-aire/>). In the first case, HERMESv3_GR is used to provide global primary aerosol emissions to the NMMB-MONARCH model, whereas in the AIRE-CDMX it is used to process the biomass burning emissions reported by GFASv1.2. Besides the two aforementioned implementations, HERMESv3_GR has been also used to perform simulations with the CALIOPE air quality forecasting system, which is based on CMAQ (<http://www.bsc.es/caliope/en/forecasts?language=en>) and in several tests using the WRF-Chem model.

## 4    Conclusions

This paper presents HERMESv3_GR, a stand-alone multiscale emission processing system that estimates gas and aerosol emissions for use in atmospheric chemistry models. HERMESv3_GR is designed to combine and process existing inventories for the generation of emission input files for global and regional air quality modelling. During the execution, emissions from different inventories, sources and specie are combined and regridded to the destination domain, and are vertically and temporally disaggregated, speciated and converted to the required format of the atmospheric chemistry model of interest. HERMESv3_GR is driven by configuration files that provide a flexible and transparent platform for the design and implementation of intercomparison and sensitivity modelling experiments.

HERMESv3_GR represents an effort of homogenizing the current available information on emission inventories and of processing them in a transparent and flexible way to produce emission outputs that can be used directly by multiple atmospheric chemistry models.

There are several features that makes HERMESv3_GR an unique emission processing system, including:

- User-defined grid and choice between different map projections: Emissions can be computed on any global or regional domain with a regular lat-lon, rotated lat-lon, mercator or lambert conformal conic projection.

- Choice between different emission inventories: the emission data library of HERMESv3_GR includes current state-of-the-art global and regional inventories that cover different sources (anthropogenic, biomass burning, volcanoes), pollutants (ozone precursor gases, acidifying gases and primary particulates) and base years (past, present and future). Moreover, country-specific scaling and masking factors defined by the user can be applied to the base inventories in order to combine and adjust them.

- Choice between different vertical, temporal and speciation profiles: HERMESv3_GR includes a dataset of profiles reported by the literature, but it also allows the user to add its own weighting factors for any pollutant sector and species. Additionally, the processing system is able to combine base inventories with gridded temporal profiles, which can be of importance for those source sectors whose temporal variation is not uniform across space (e.g. residential combustion emissions driven by temperature).

- Choice between different atmospheric chemistry model: The generated emission files can be used as input for the CMAQ, WRF-CHEM and NMMB-MONARCH chemical transport models.

- Choice between different chemical mechanisms: base pollutants can be mapped to several gas-phase and aerosol chemical mechanism, including CB05, CB05e51, RADM2, AERO5, AERO6 and MADE/SORGAM. All these mechanisms are widely used in the air quality modelling community.

- Parallel implementation: The emission core module of HERMESv3_GR is parallelized using a domain decomposition strategy, which allows decreasing the execution time and memory consumption of the model. This feature can be of importance when using the processing system in operational air quality forecasting systems, for which the simulations need to be completed within the required time constraints.

Several emission outputs obtained with HERMESv3_GR are provided in this paper to illustrate its potential. The software is implemented within NMMB-MONARCH and CMAQ in the framework of the ICAP multi-model ensemble and the AIRE-CDMX air quality forecasting system for Mexico City, respectively.

It is worth noting that despite providing a flexible and simplified framework for the processing of emissions, user should have a clear knowledge of the original inventories when using HERMESv3_GR. Combining parts from different inventories could lead to substantial errors (e.g. double counting) because the definition of what is included or excluded in certain sectors and/or inventories can differ significantly (e.g. agricultural waste burning emissions are sometimes included under the agriculture source sector and sometimes excluded). It is therefore recommended that users carefully check the original descriptions of each inventory before using them. With the aim of facilitating this task, the HERMESv3_GR wiki (see Sect. 5) includes a section with a general description of each inventory and links to the official references.

Future work will consider the expansion of the emission data library to include regional inventories of regions such as Asia or America, emission datasets that are currently being developed in the framework of the Copernicus Atmosphere Monitoring

Service (CAMS), as well as datasets that report emissions of greenhouse gases, so that HERMESv3_GR can also serve as input for climate modelling. Other efforts will focus on the implementation of a functionality to handle the remapping of emissions to unstructured destination grids (e.g. octahedral grid), which are starting to be widely used in global models due to their computational efficiency and effective resolution, as well as on the inclusion of sector-dependent spatial proxies during

5    the remapping process and the improvement of the scalability of the writing function.

## 5    Code availability

The HERMESv3_GR code package, pre-processing functions to homogenize the emission inventories (listed in Table 1), sample configuration and ancillary input files (vertical, temporal and speciation profiles) and a test case data are available at the following gitlab repository: https://earth.bsc.es/gitlab/es/hermesv3_gr. A wiki of the processing system with further

10   instructions is also included in the gitlab repository, as well as the links and references for downloading and citing the original gridded emission inventories that HERMESv3_GR can process. The required libraries need to be installed by the user in the computer infrastructure where the processing system is planned to be run.

# 6 Appendices

## Appendix A: HERMESv3_GR general configuration file (*hermes.conf*)

| Parameters and examples | Description and comments |
|---|---|
| [GENERAL] | |
| log_level = 3 | Defines the logging level, which is associated to the amount of information that will appear in the log file. The options are 1, 2 or 3 (recommended for debugging) |
| input_dir = /gpfs/projects/HERMESv3/IN | Defines the general input directory of the model |
| data_path = /gpfs/scratch/data/ | Defines the common directory path where all the homogenised emission inventories used by HERMESv3_GR are stored. The complete path to each specific emission inventory file is specified in the emission inventory configuration file |
| output_dir = /gpfs/projects/HERMESv3/OUT | Defines the directory where the output emission files will be stored |
| output_name = HERMESv3_<date>.nc | Name of the output emission file. The string <date> is automatically replaced by the starting date of the simulation day. The complete path to the output file is the combination of output_dir and output_name. |
| start_date = 2010/01/01 00:00:00 | Starting date of the simulation (in UTC). Date formats accepted by HERMESv3_GR include: <ul><li>YYYYMMDD: 20150101</li><li>YYYYMMDDhh: 2015010100</li><li>YYYYYMMDD.hh: 20150101.00</li><li>YYYY/MM/DD: 2015/01/01</li><li>YYYY/MM/DD_hh: 2015/01/01_00</li><li>YYYY/MM/DD_hh:mm:ss: 2015/01/01_00:00:00</li><li>YYYY/MM/DD hh:mm:ss: 2015/01/01 00:00:00</li><li>YYYY-MM-DD_hh: 2015-01-01_00</li><li>YYYY-MM-DD_hh:mm:ss: 2015-01-01_00:00:00</li><li>YYYY-MM-DD hh:mm:ss: 2015-01-01 00:00:00</li></ul> |
| end_date = 2010/01/02 00:00:00 | [OPTIONAL]Ending date of the simulation (in UTC). If it is not set then end_date = start_date. |
| output_timestep_type = hourly | Temporal resolution of the output file. The options are: <ul><li>Hourly</li><li>Daily</li><li>Monthly</li></ul> |
| output_timestep_num = 24 | Number of time steps to simulate |
| output_timestep_freq = 1 | Frequency between time steps |
| [DOMAIN] | |
| output_model = CMAQ | Defines the format of the output emission file as a function of the atmospheric chemistry model conventions. Current options are: <ul><li>MONARCH</li><li>CMAQ</li><li>WRF_CHEM</li></ul> |

| | |
|---|---|
| output_attributes = <input_dir>/data/cmaq_global_attributes.csv | Path to the file that contains the global attributes that need to be included in the output NetCDF file according to the corresponding chemical transport model |
| domain_type= lcc | Defines the grid projection on which the emission fields will be generated. Options are:<br>• global: regular lat-lon grid<br>• rotated: rotated lat-lon grid<br>• lcc: lambert conformal conic grid<br>• mercator: mercator grid |
| vertical_description = <input_dir>/data/profiles/vertical/vert.csv | Path to the file that contains the vertical description of the desired output |
| aux_files_path = <input_dir>/data/aux_files/<domain_type>_<res> | Path to the directory where the necessary auxiliary files (e.g. timezones file) will be created if they do not exist. If they already exist, HERMESv3_GR will just read them |
| # if domain_type == global:<br>    inc_lat = 0.5<br>    inc_lon = 0.703125 | Parameters that define a global regular lat-lon grid:<br>• inc_lat: Latitudinal grid resolution (degrees)<br>• inc_lon: Longitudinal grid resolution (degrees). |
| # if domain_type == rotated:<br>    centre_lat = 35<br>    centre_lon = 20<br>    west_boundary = -51<br>    south_boundary = -35<br>    inc_rlat = 0.1<br>    inc_rlon = 0.1 | Parameters that define a regional rotated lat-lon grid:<br>• centre_lat = Central geographic latitude of the grid (non-rotated degrees).<br>• centre_lon = Central geographic longitude of grid (non-rotated degrees, positive east).<br>• west_boundary = Grid's western boundary from center point (rotated degrees).<br>• south_boundary = Grid's southern boundary from center point (rotated degrees).<br>• inc_rlat = Latitudinal grid resolution (rotated degrees).<br>• inc_rlon = Longitudinal grid resolution (rotated degrees). |
| # if domain_type == lcc:<br>    lat_1 = 37<br>    lat_2 = 43<br>    lon_0 = -3<br>    lat_0 = 40<br>    nx = 278<br>    ny = 298<br>    inc_x = 1000<br>    inc_y = 1000<br>    x_0 = 253151.59375<br>    y_0 = 43862.90625 | Parameters that define a regional lambert conformal conic grid:<br>• lat_1 = Standard parallel 1 (in degrees).<br>• lat_2 = Standard parallel 2 (in degrees).<br>• lon_0 = Longitude of the central meridian (in degrees).<br>• lat_0 = Latitude of the origin of the projection (in degrees).<br>• nx = Number of grid columns.<br>• ny = Number of grid rows.<br>• inc_x = X-coordinate cell dimension (in meters).<br>• inc_y = Y-coordinate cell dimension (in meters).<br>• x_0 = X-coordinate origin of grid (in meters).<br>• y_0 = Y-coordinate origin of grid (in meters). |

| | |
|---|---|
| # if domain_type == mercator:<br>   lat_ts = -2.84<br>   lon_0 = -79.16<br>   nx = 99<br>   ny = 81<br>   inc_x = 1000<br>   inc_y = 1000<br>   x_0 = -49500.13899<br>   y_0 = -355986.692 | Parameters that define a regional lamber conformal conic grid:<br>• lat_ts = Latitude of true scale (in degrees).<br>• lon_0 = Longitude of projection center (in degrees).<br>• nx = Number of grid columns.<br>• ny = Number of grid rows.<br>• inc_x = X-coordinate cell dimension (in meters).<br>• inc_y = Y-coordinate cell dimension (in meters).<br>• x_0 = X-coordinate origin of grid (in meters).<br>• y_0 = Y-coordinate origin of grid (in meters). |
| [EMISSION_INVENTORY_CONFIGURATION] | |
| cross_table = <input_dir>/conf/EI_conf.csv | Defines the path to the emission inventory configuration file |
| [EMISSION_INVENTORY_PROFILES] | |
| p_vertical =<br><input_dir>/data/profiles/vertical/vert_prof.csv | Defines the path to the file that contains the vertical profiles. |
| p_month =<br><input_dir>/data/profiles/temporal/month.csv | Defines the path to the file that contains the monthly temporal profiles. |
| p_day = <input_dir>/data/profiles/temporal/day.csv | Defines the path to the file that contains the daily temporal profiles. |
| p_hour = <input_dir>/data/profiles/temporal/hour.csv | Defines the path to the file that contains the hourly temporal profiles. |
| p_speciation =<br><input_dir>/data/profiles/speciation/spec_cb05aero5.csv | Defines the path to the file that contains the speciation profiles. |
| molecular_weights =<br><input_dir>/data/profiles/speciation/MW.csv | Defines the path to the file that contains the molecular weights of the input pollutant species. |
| world_info =<br><input_dir>/data/profiles/temporal/tz_iso3166.csv | Defines the path to the file that contains the mapping between worldwide time zones and country ISO3 codes. This file is used to create the time zone grid for the temporal disaggregation of the emissions. |

## 7    Author contribution

Marc Guevara conceived and coordinated the development of HERMESv3_GR, prepared all the input databases (vertical, temporal and speciation profiles) and selected the inventories to be included in the emission data library. Manuel Porquet Pardina helped preparing the input databases and performing software tests. Carles Tena developed the HERMESv3_GR code and run the experiments to test the performance of the parallel implementation. Oriol Jorba helped conceiving HERMESv3_GR and its implementation within the NMMB-MONARCH model. Carlos Pérez García-Pando helped conceiving HERMESv3_GR and supervised the work. Marc Guevara prepared the manuscript with contributions from all co-authors.

## 8    Acknowledgements

The research leading to these results has received funding from the Ministerio de Economía y Competitividad (MINECO) as part of the PAISA project CGL2016-75725-R and the NUTRIENT project CGL2017-88911-R. The authors acknowledge PRACE for awarding access to Marenostrum4 based in Spain at the Barcelona Supercomputing Center through the Tier-0 HHRNTCP and Tier-0 EEDMC projects. Carlos Pérez García-Pando acknowledges long-term support from the AXA Research Fund, as well as the support received through the Ramón y Cajal programme (grant RYC-2015-18690) of the Spanish Ministry of Economy and Competitiveness. The authors would also like to thank the two anonymous referees for their thorough comments, which helped improve the quality of the paper.

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

**Table 1: Summary of the input emission inventories currently available in the HERMESv3_GR library.**

| Name | Sources | Spatial resolution / coverage | Temporal resolution / coverage | Pollutant species | Re |
|---|---|---|---|---|---|
| EDGARv4.3.2_AP | Anthropogenic | Global (0.1x0.1) | Annual (1970 – 2012) Monthly (2010) | NOx, CO, SO2, NH3, NMVOC, PM10, PM2.5, OC, BC | Cripa e |
| EDGARv4.3.2_VOC | Anthropogenic | Global (0.1x0.1) | Annual (1970 – 2012) Monthly (2010) | GEIA 25 NMVOC groups | Huang |
| CEDS | Anthropogenic | Global (0.5x0.5) | Monthly (1851 – 2014) | NOx, CO, SO2, NH3, NMVOC (and the GEIA 25 NMVOC groups), OC, BC | Hoesly |
| ECLIPSEv5.a | Anthropogenic | Global (0.5x0.5) | Monthly (1990 - 2050) | NOx, CO, SO2, NH3, NMVOC, PM10, PM2.5, OC, BC | Klimont |
| HTAPv2.2 | Anthropogenic | Global (0.1x0.1) | Monthly (2008 and 2010) | NOx, CO, SO2, NH3, NMVOC (and the GEIA 25 NMVOC groups [1]), PM10, PM2.5, OC, BC | Janssens-( |
| GFASv1.2 | Biomass burning | Global (0.1x0.1) | Daily (2012-present) | NOx, CO, SO2, NH3, PM2.5, OC, BC, CH3OH, C2H5OH, C3H8, C2H4, C3H6, C5H8, terpenes, hi alkenes, hi alkanes, CH2O, C2H4O, C3H6O, C2H6S, C2H6, C7H8, C6H6, C8H10, C4H8, C5H10, C6H12, C8H16, C4H10, C5H12, C6H14, C7H16 | Kaiser |
| Carn_etal | Volcanoes (degassing) | Point sources (lat-lon) | Annual (2005 – 2015) | SO2 | Carn e |
| Wiedinmyer_etal | Open air trash burning | Global (0.1x0.1) | Annual (2010) | NOx, CO, SO2, NH3, PM10, PM2.5, OC, BC, C2H2, C2H4, C3H6, C6H6, CH2O, CH3COOH, CH3OH, HCL | Wiedinmy |
| TNO_MACC-iii | Anthropogenic | Regional (0.0625x0.125) | Annual (2000 – 2011) | NOx, CO, SO2, NH3, NMVOC (and the GEIA 25 NMVOC groups [2]), PM10, PM2.5, OC, BC | Kuenen |
| EMEP | Anthropogenic | Regional (0.1x0.1) | Annual (2000 – 2016) | NOx, CO, SO2, NH3, NMVOC, PM10, PM2.5 | Mareckov |

[1] Based on the NMVOCs breakdown ratios generated for the RETRO project (Schultz et al., 2007)
[2] Based on the NMVOCs breakdown ratios generated for the AQMEII modelling exercise (Pouliot et al., 2015)

**Table 2: Example of speciation profiles included in HERMESv3_GR for mapping the GFASv1.2 emissions to CB05 and AERO5 and the CEDS road transport emissions to RADM2 and MADE/SOGARM chemical mechanisms.**

| GFASv1.2 CB05 + AERO5 speciation profile | | CEDS road transport RADM2 + MADE/SORGAM speciation profile | |
|---|---|---|---|
| specie | expression | specie | expression |
| NO | 0.72*nox_no [1] | NO | 0.84*nox_no2 [2] |
| NO2 | 0.18*nox_no [1] | NO2 | 0.16*nox_no2 [2] |
| HONO | 0.1*nox_no [1] | CO | co |
| CO | co | SO2 | so2 |
| SO2 | so2 | NH3 | nh3 |
| NH3 | nh3 | ALD | voc22 |
| ALD2 | c2h4o | ETH | voc02 |
| ALDX | 0 | HC3 | 0.95*voc01+voc03+voc04+0.4*voc09+0.69*voc18+voc20 |
| BENZENE | c6h6 | HC5 | 0.05*voc01+voc05+0.43*voc06+0.31*voc18 |
| ETH | c2h4 | HC8 | 0.57*voc06+voc17+voc19 |
| ETHA | c2h6 | HCHO | voc21 |
| ETOH | c2h5oh | ISO | voc10 |
| FORM | 0 | KET | voc23 |
| IOLE | 0.5*hialkenes | OL2 | voc07 |
| ISOP | c5h8 | OLI | voc11+voc12 |
| MEOH | ch3oh | OLT | voc08 |
| OLE | c8h16+c5h10+c3h6+c4h8+c6h12+0.5*hialkanes | ORA1 | 0.44*voc24 |
| PAR | 4*c4h10+6*c6h14+5*hialkanes+6*c8h16+3*c5h10+c3h6+3*c3h6o+2*c4h8+7*c7h16+4*c6h12+hialkenes+5*c5h12+1.5*c3h8 | ORA2 | 0.56*voc24 |
| SESQ | 0 | TOL | 0.293*voc13+voc14 |
| TERP | terpenes | XYL | voc16+voc17 |
| TOL | ch2o+c7h8 | PM_10 | 0 |
| XYL | c8h10 | PM25J | 0 |
| DMS | c2h6s | PM25I | 0 |
| HCL | 0 | ECJ | bc*0.8 |
| POA | 1.8*oc | ECI | bc*0.2 |
| PEC | bc | ORGJ | oc*0.8 |
| PNO3 | 0 | ORGI | oc*0.2 |
| PSO4 | 0 | NO3J | 0 |
| PMFINE | pm25-oc-bc | NO3I | 0 |
| PMC | 0 | SO4J | 0 |
| SULF | 0 | SO4I | 0 |

[1] GFASv1.2 $NO_x$ emissions are reported as NO
[2] CEDS $NO_x$ emissions are reported as $NO_2$

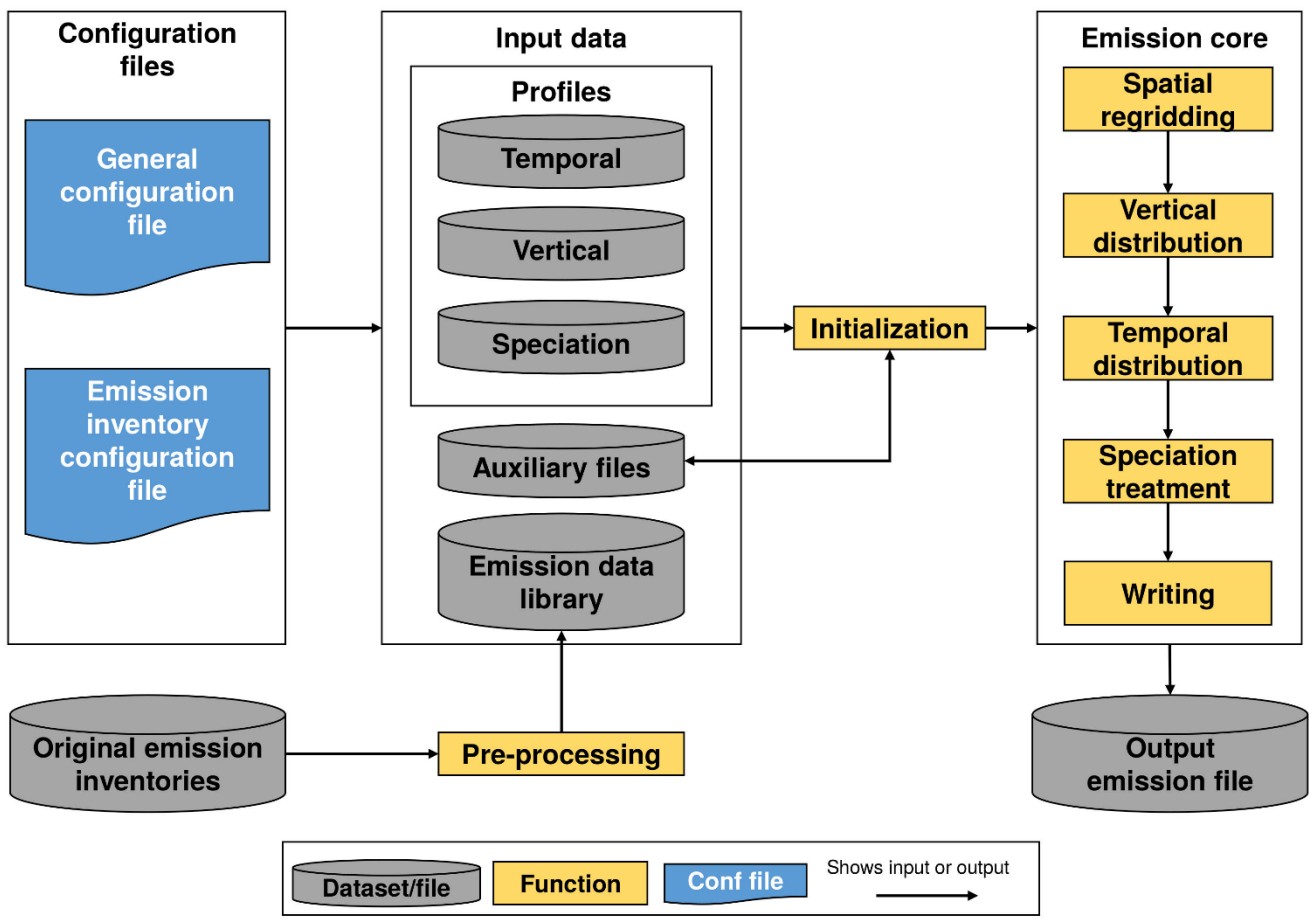

**Figure 1: Schematic representation of the general structure of HERMESv3_GR**

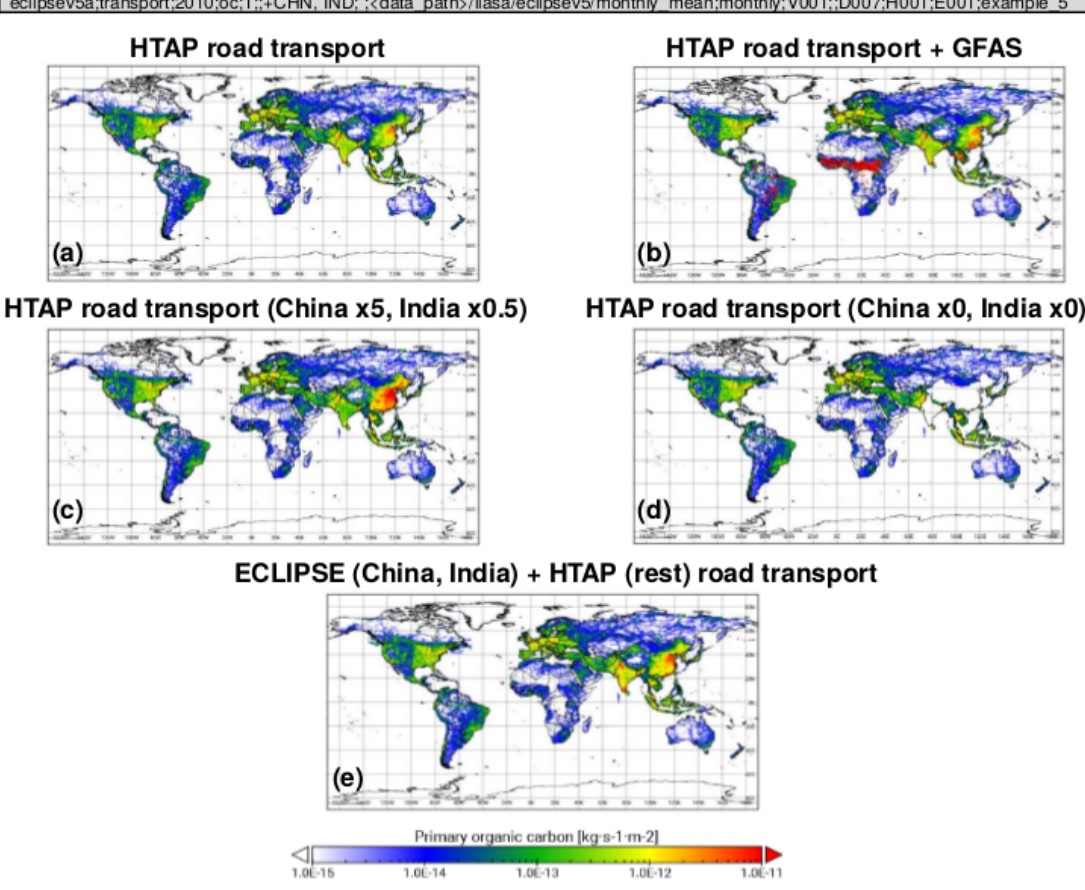

**Figure 2: Examples of organic carbon global emission outputs regridded onto a 0.5x0.7 degree global regular lat-lon domain obtained with HERMESv3_GR using five different versions of the emission inventory configuration file: HTAP road transport (a), HTAP road transport + GFAS (b), HTAP road transport with scaling factors over China (5) and India (0.5) (c), HTAP road transport masking out China and India (d) and ECLIPSE road transport (China and India) + HTAP road transport (rest of countries) (e). The corresponding emission inventory configuration files used in each example are shown at the top.**

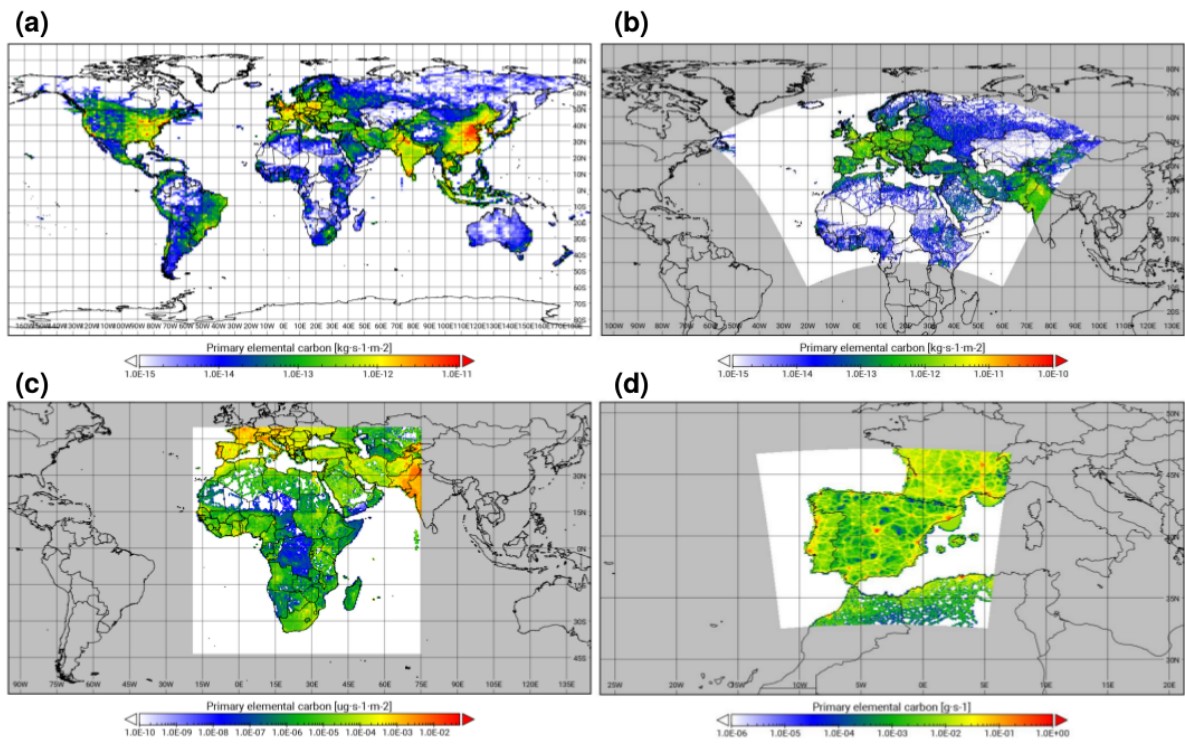

**Figure 3: Examples of the HTAPv2.2 black carbon transport emissions regridded onto a: 1x1.4 degree global regular lat-lon domain (a), 0.1x0.1 degree rotated lat-lon domain (b), 50x50 km mercator grid (c) and 4x4 km lambert conformal conic grid (d). All maps are displayed in an Equirectangular projection.**

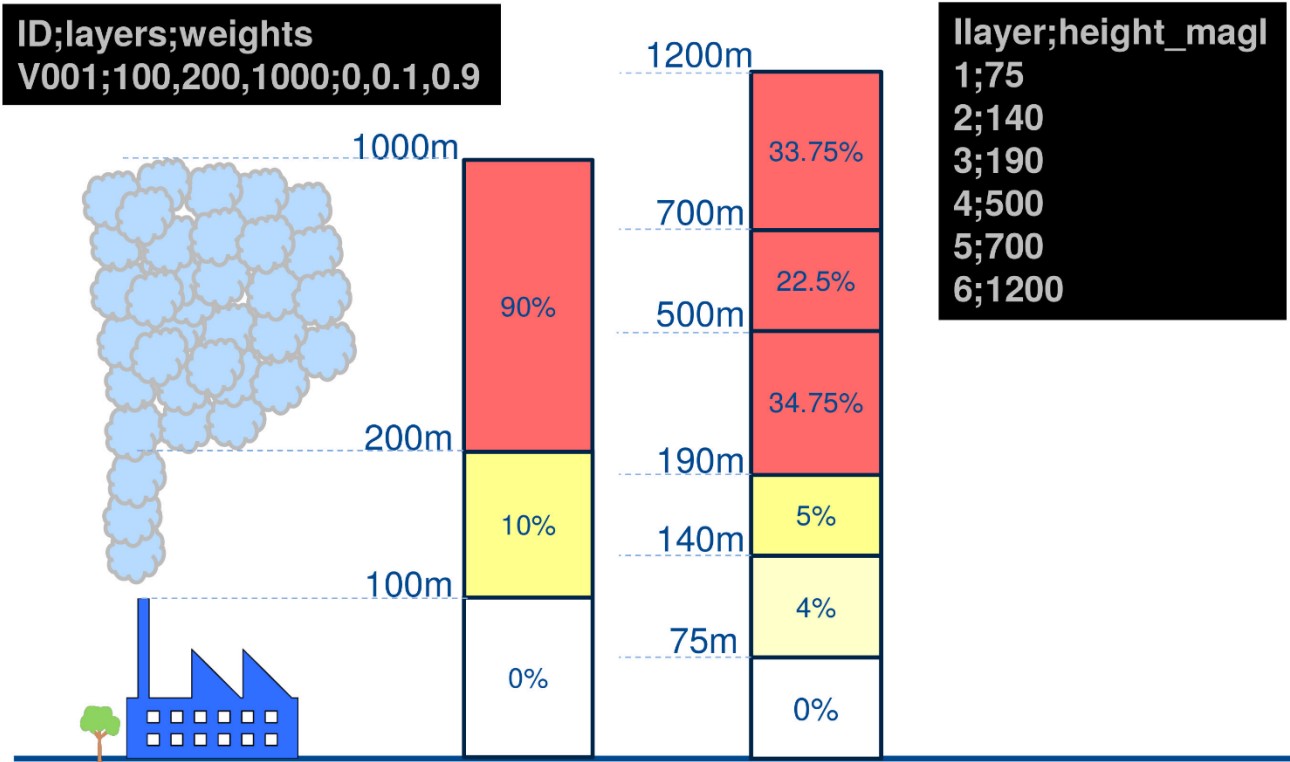

**Figure 4: Schematic representation of the emission vertical distribution process implemented within HERMESv3_GR. Left side shows an example of a vertical profile description ("V001"), which allocates 10% of emissions between 100 and 200 m.a.g.l. and the remaining 90% between 200 and 1000 ma.g.l.. Right side shows an example of the vertical description of the domain. Original vertical weights are interpolated to the model vertical layers according to their thickness.**

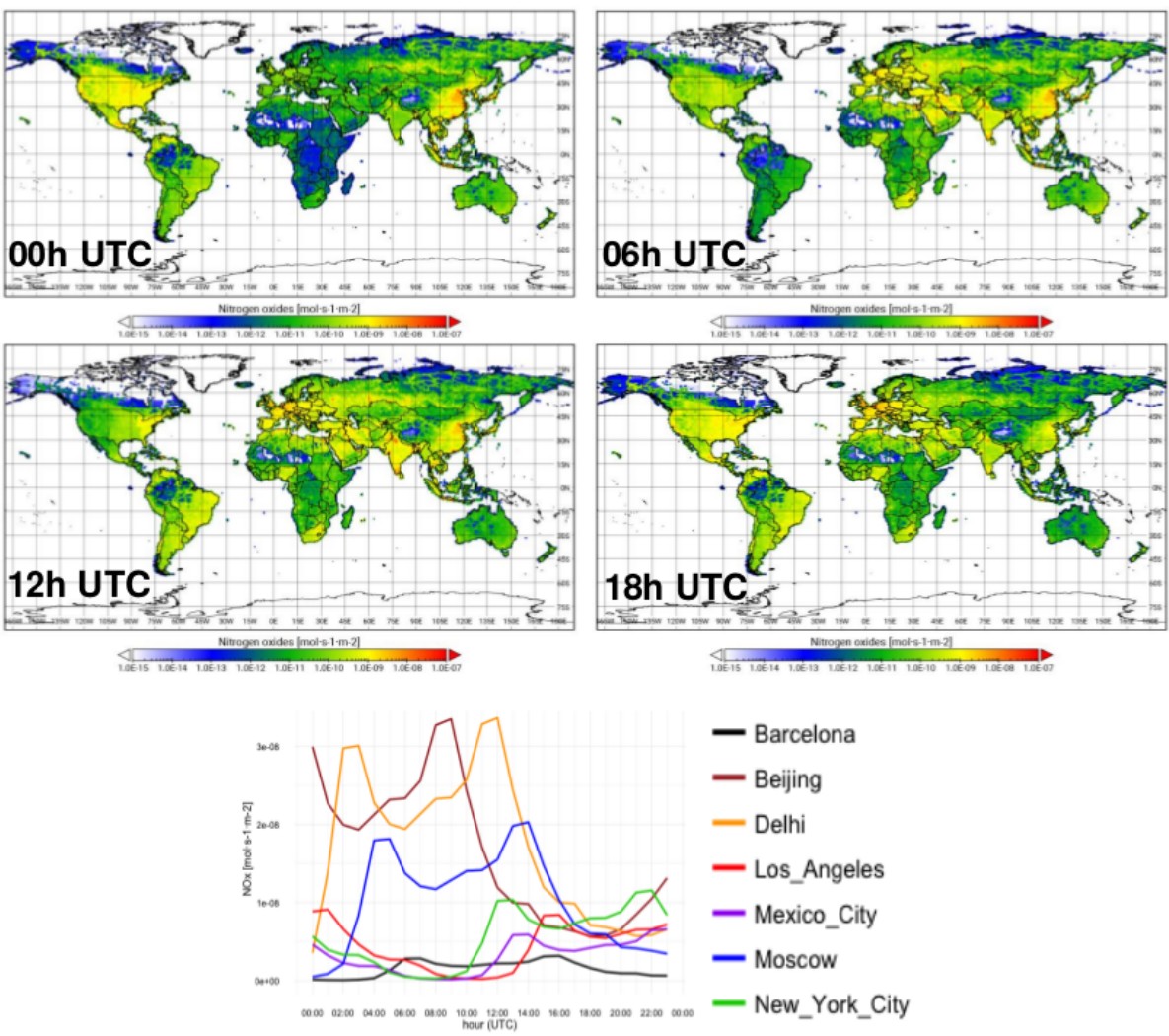

**Figure 5: Global hourly NOx transport emissions [mol·s-1·m-2] derived from ECLIPSEv5a at 00:00h (a), 06:00h (b), 12:00h (c) and 18:00h (d) UTC and the diurnal evolution estimated in the grid cells where different global cities are located (e).**

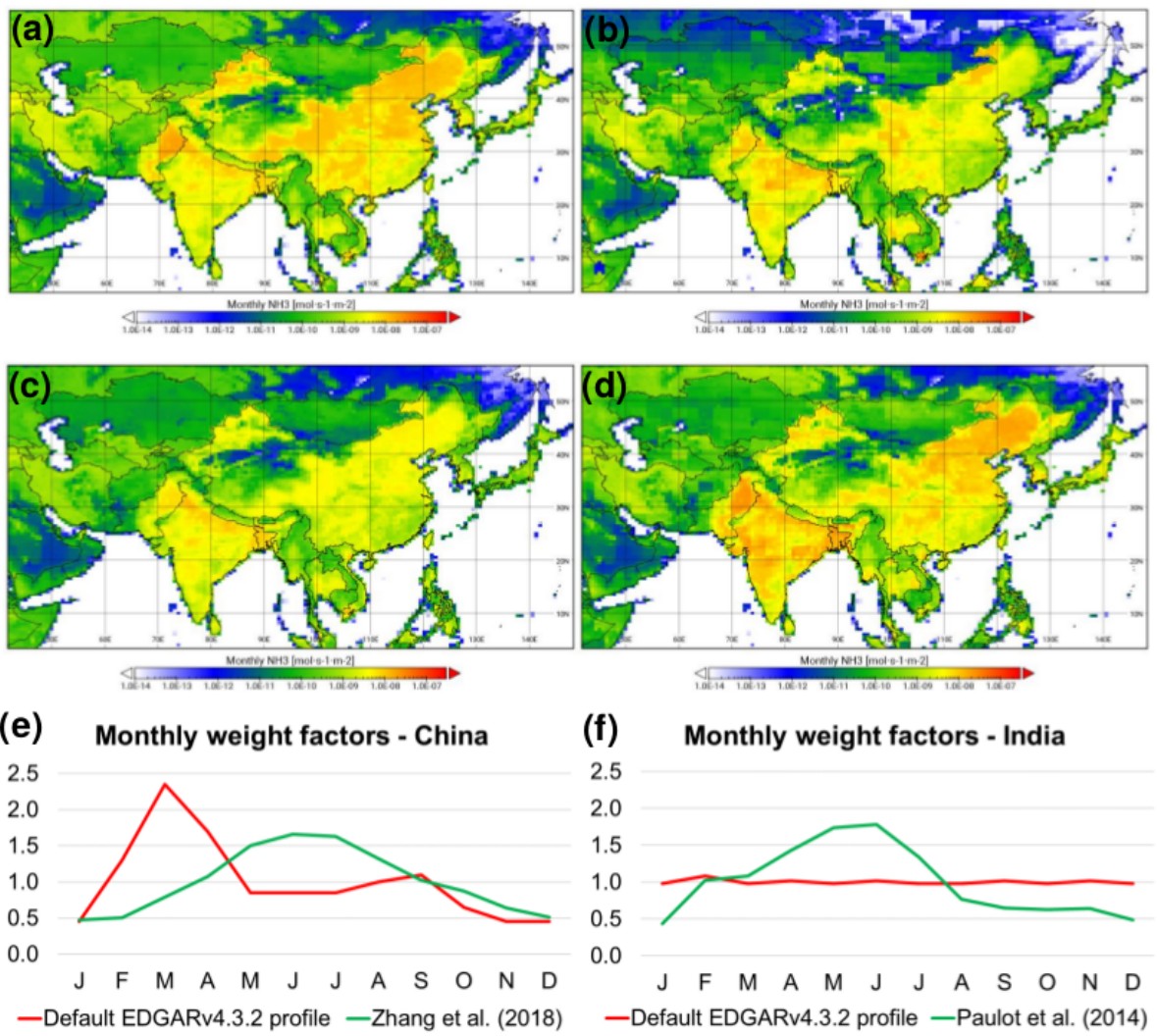

**Figure 6: Monthly NH3 agricultural soil emissions [mol·s-1·m-2] estimated with HERMESv3_GR in East Asia (0.5x0.7 degree) for March and June using the default temporal profiles reported by EDGARv4.3.2 (a and c) and a gridded temporal profile derived from the works of Paulot et al. (2014) and Zhang et al. (2018) (b and d), and monthly weight factors obtained in China and India for each case (e and f).**

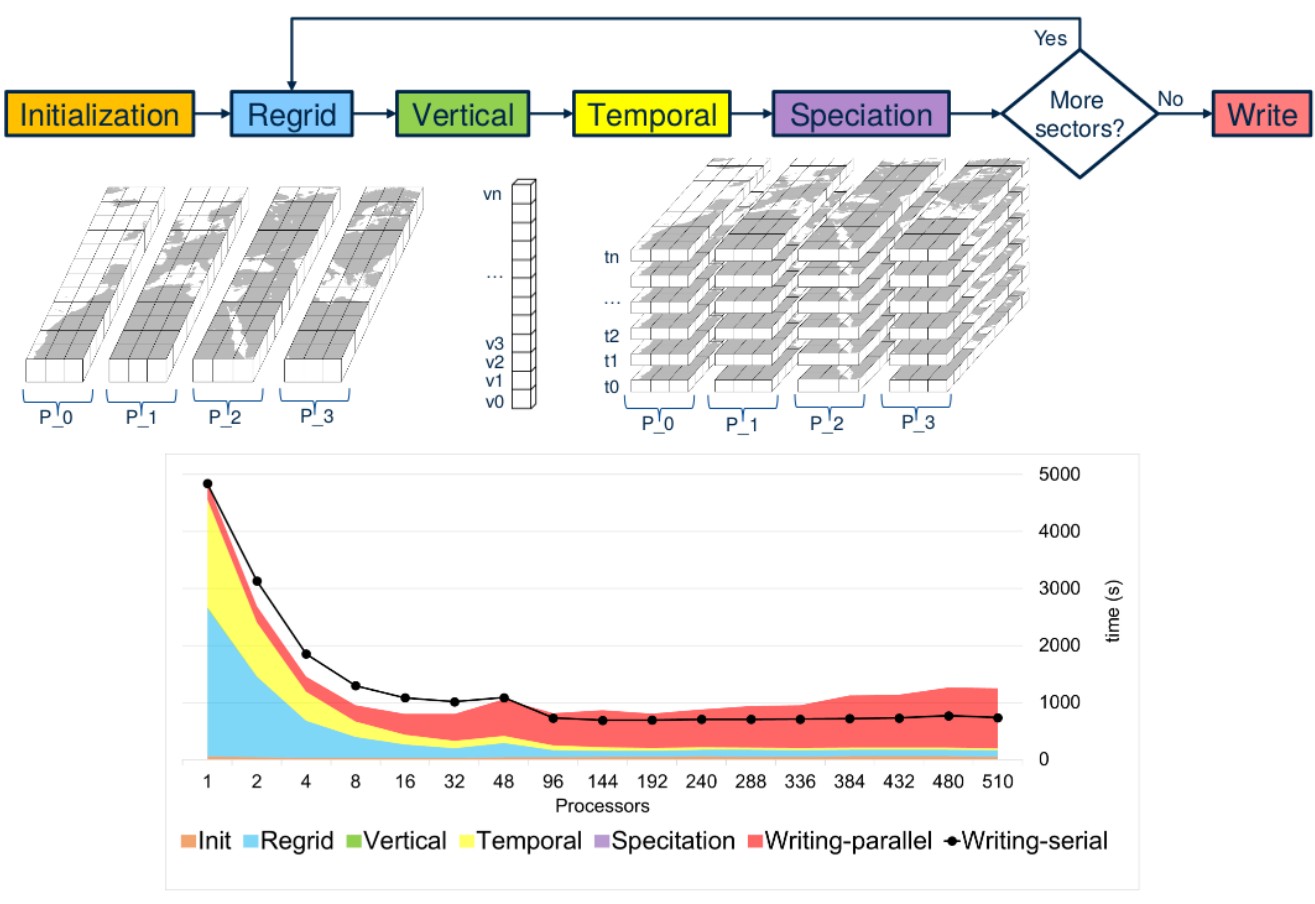

**Figure 7: Schematic representation of the parallelization of the emission core module of HERMESv3_GR (top) and computational times obtained for each functionality (regrid, vertical, temporal, speciation and writing-parallel/writing-serial) for the scalability test performed (bottom). The destination working domain is divided into vertical sections, according to the number of processors to be used (P_0, P_1, …). Vertical (v0, v1, …) and temporal (t0, t1, …) weight factors are applied to each section in order to transform the 2D arrays (longitude, latitude) into 4D arrays (time, vertical layer, longitude, latitude).**