# Peer review of "HERMESv3, a stand-alone multiscale atmospheric emission modelling framework - Part 1: global and regional module."

_Geoscientific Model Development, 2018_

## Referee Comment (RC1) · Anonymous Referee #1 · 5 Feb 2019

The paper describes the structure of HERMESv3, an open source parallel tool that can be use to create emission input files for various air quality. The strength of HERMESv3 is without a doubt in its ability to process various databases for various air quality models and its flexibility as users can easily choose different parameters and ways to create the emission files. It is therefore certain that HERMESv3 will be widely use by modelers, especially if the list of models and mechanisms compatible increases in the future. The paper is well structured although it is sometimes lacking details. Therefore, the paper should be revised according to the following comments before being published.

[Figure]

Major comments:

One of my concern is that the tool does not take into account some meteorological parameters as it may prevent the code from being used by some air quality models for two reasons. First, some air quality models do not have a constant vertical grid but uses sigma levels. For these models, altitude of the different vertical layers of the model will change with time and space. From what I understand of HERMESv3, the model should not be able to directly distribute the emissions on those vertical grids. Can HERMESv3 somehow treat this specific case or does it mean that the models have to be adapted to read the emissions from HERMESv3? Second, some methods have been developed to temporalize (or even spatialize) the emissions from several sources (for example residential wood burning, agriculture) and such methods are used by some models. I understand it would have been difficult to do in a first approach, but it may be useful to indicate if such methods could be implemented into HERMES.

The figures should be improved. The scale of the maps should be revised (as the maps are almost entirely blue) to improve the readability and increase the number of details. I would recommend using a log scale to avoid showing only high values, and to not color areas without emissions. In figure 2, the order of the columns (ei, sector, ref_year, active, factor_mask, regrid_mask, pollutants) does not correspond to the order in the text (ei, sector, ref_year, pollutants, active, factor_mask, regrid_mask), making the text a bit difficult to follow. Moreover, the examples in figure 2 are difficult to understand without referring to the text. Moreover, I wonder if there could be a mistake in example 1, as the pollutants "nox_no2" and "co" are written in the columns whereas the caption of the figure and the text refer to OC emissions (not CO).

P8 l26: What does the authors means by first-order conservative? The method of Hill et al. (2004) should be better explained. From what I understand, HERMES does not use the landuse and only distribute homogeneously the emissions and therefore could distribute land emissions over seas or distribute agricultural emissions onto cities.

[Figure]

P8 l31-32: the authors use a gridding country mask to allocate emissions to a specific country. How are separated the emissions when there are several countries into a cell? Some inventories (like the EMEP inventory) directly provide the information of the emitting country. In that case, using a country mask is not useful. Is the information of emitting country use when given?

P9 l4-10: several methods are presented to distribute emissions onto vertical layers. A discussion on the comparison of the methods, with the strength and weaknesses of each methods, would be appreciated.

Speciation mapping: This section lacks details and several elements seem weird. In this state, it gives the impression that the speciation is not treated appropriately. For NMVOCs, I don't understand how it is possible to convert from mass to moles before using the speciation. You would need to know the speciation to compute the mean molar masses of NMCOVs. For NOx, I guess that you use the molar mass of NO2 if the NOx emissions are given as NO2 equivalent. Some explanations on Table 2 are needed. I don't understand why: - NO = nox_no2 and NO2=0.18*nox_no - TOL=0.293*voc13 (said to be benzene) +voc14 (said to be toluene) while there is a separate benzene species (and why 0.293). A similar question can be asked for almost species. - POA=3*oc (if it is to convert OC emissions into OM emissions, a factor 3 is very high and very unlikely) - EC=5.9*bc (it seems like that the emissions are artificially increase by a factor 5.9) - PMfine = 3.3*pm25-3*oc-5.9*bc (it seems like the mass of PM is artificially increase by a factor 3.3)

Writing module: as Figure 7 shows the time for writing increase with the number of processors used. As the authors said, the writing function does not scale properly, probably due to the NetCDF 4 library. Did the authors try to write (if possible) the results with only one processor or the use a specific library (like pnetcdf) for parallel writing?

Minor comments:

P2 l4: the authors should add a few words on why the global and regional inventories are too imprecise for urban scale modeling

P9 l11: If you transform a 0.1°x0.1° inventory into 1°x1.4° emissions, it is not technically an interpolation. I would not use the word interpolation in the text and only use the word regridding.

P9 l22: a.g.l is not defined

P12 l5: ":" instead of "Table 2" at the beginning of the line 6

P15 l30: "which are starting to be widely used in global models"

––––––––––––––––––––––––––––––––

---

## Short Comment (SC1) · 4 Mar 2019

It is an important paper about emissions. Some questions after reading it are the following:

1) It is not clear what year the emissions are valid.

2) There is not shown a methodology validation for Eq 1. If someone used the emission output results for every single hour and added for a year, in the same selected area, he should have the initial emission value of that area. Is it done before?

3) Figure 5 page 33 Emission from Beijing and Delhi looks the same. Is it true?

---

## Referee Comment (RC2) · Anonymous Referee #2 · 15 Mar 2019

Review: gmd-2018-324 Title: HERMESv3, a stand-alone multiscale atmospheric emission modelling framework - Part 1: global and regional module Author(s): Marc Guevara et al.

The paper describes an open source system to process various emission datasets is a flexible manner allowing for changes in projections, scales and making combinations of different inventories. Moreover it provides options for applying different temporal or emission height profiles to generate model-ready emissions input. One of the nice things is that it will allow modelers to relatively easy do sensitivity tests by the ability to scale and/or quickly combine various sets. I do think there is some risk in this,

in the sense that people who use it may think that everything is compatible and you can "shop" until you find what you need but in the end this is more a concern than a comment on the paper. The paper is well written and clear. In my opinion it is a good contribution for GMD and I only have minor comments which should be taken into account before accepting the paper.

Abstract: please remove "highly" in l 10. It is customizable but highly is an undefined property. What you may find low, someone else may find high and vice versa. This occurs at various places.

In the introduction P2 L 18 it is stated that "A potential remedy for the latter is to combine different inventories and apply adjustment factors in order to improve the representativeness of the emission data....." This should be a bit better explained and possibly also discussed further in the paper. What does improving the representativeness mean? It is important to acknowledge that we should not work towards (and the system is not intended for) having only one totally harmonized inventory. Like models, inventories work from different assumptions with different data and solutions. Having independent datasets is crucial from a science perspective.

P2 l25 I suggest to replace "quality" with resolution – the quality may be good for a global product but not for a regional product.

P3 l4 "highly" – see previous comment

P6 l 6-7 does the user provide data? Or the data provider? I assume there can be users who do not provide data?

P8 l1-3 – This possible explanation should be removed. As it is not further documented it remains speculation and does not belong in this paper. Furthermore, for making comparisons between a certain emission category from different inventories one should not use maps but the emission data by sector.

P 12 l 6 – reference to Table 2 is missing at the start of the sentence.

P12 l 14-15 – please check if sentence is correct it sort of says that NO is mapped to NO2 but maybe I misunderstand.

P15 l 12 "and temperature" is not correct maybe you mean "driven by temperature". The sentence now implies that temperature is a pollutant sector. Also pollutant sector should be source sector.

P15 l22 remove "–"

P15 l 25 work not works

P15 l 30 widely USED in

Figures: At least when printed the maps are not very clear and while they only serve as an illustration it seems the legend is not well chosen. It would be better to show more gradients.

Finally in the conclusions it should be considered to make disclaimer or statement that the system PROCESSES emissions data, it does not make them better. Users should always remain aware that combining parts from different inventories can also lead to substantial errors because the definition what is included or excluded in certain sectors and/or inventories can differ substantially. A notorious example is e.g. agricultural waste burning which is sometimes included under agriculture sometimes excluded (and than given under waste, or not at all as it is assumed it comes from the Fire emission inventories). So combining apples and oranges without going to the original descriptions of what is included should be avoided. In the end this is the responsibility of the user but a word of warning is warranted.

---

## Author Comment (AC1) · 10 Apr 2019

Thank you for the opportunity to revise the manuscript. The comments of the reviewers are indicated point-by-point in the attached pdf file. We explain how we have carefully addressed each of them (our answers in blue text). In the last page we also reply to the short comments from the scientific community.

Please also note the supplement to this comment:
https://www.geosci-model-dev-discuss.net/gmd-2018-324/gmd-2018-324-AC1-supplement.pdf

---

## Author Response (AR1)

Thank you for the opportunity to revise the manuscript. The comments of the reviewers are indicated point-by-point in the following text. We explain how we have carefully addressed each of them (our answers in blue text). Modifications and new sections are highlighted with track changes in the revised version of the manuscript.

**Anonymous Referee #1**

The paper describes the structure of HERMESv3, an open source parallel tool that can be used to create emission input files for various air quality. The strength of HERMESv3 is without a doubt in its ability to process various databases for various air

10   quality models and its flexibility as users can easily choose different parameters and ways to create the emission files. It is therefore certain that HERMESv3 will be widely used by modelers, especially if the list of models and mechanisms compatible increases in the future. The paper is well structured although it is sometimes lacking details. Therefore, the paper should be revised according to the following comments before being published.

We appreciate the appraisal of Reviewer #1 and his/her thorough comments, which helped improve the quality of the paper.

15   We indeed have the plan to extended HERMESv3_GR to other European atmospheric chemistry models (e.g. CHIMERE, LOTOS-EUROS).

Major comments:

One of my concern is that the tool does not take into account some meteorological parameters as it may prevent the code from being used by some air quality models for two reasons. First, some air quality models do not have a constant vertical grid but

20   uses sigma levels. For these models, altitude of the different vertical layers of the model will change with time and space. From what I understand of HERMESv3, the model should not be able to directly distribute the emissions on those vertical grids. Can HERMESv3 somehow treat this specific case or does it mean that the models have to be adapted to read the emissions from HERMESv3?

Response to Reviewer#1 comment No. 1: HERMESv3_GR is currently designed as an off-line model and cannot use or take

25   into account dynamic environmental variables (e.g. height of vertical layers) provided by atmospheric chemistry models. The tool generates output emission files that can be directly read by several atmospheric chemistry models (i.e. NMMB-MONARCH, CMAQ, WRF-Chem) but does not interact with them directly during the processing of the emissions.

Consequently, HERMESv3_GR cannot distribute the emissions to the exact sigma levels of the model (which varies in time and space, with temporal resolution in some cases of some seconds) but to a set of fixed vertical levels that are close to them.

30   In this sense, the user is able to define the vertical description that thinks is more suitable to the corresponding atmospheric chemistry model.

For instance, for the definition of the 48 vertical levels in a 1.4x1.0 degree global domain of the NMMB-MONARCH model (which is an atmospheric chemistry model that uses a mass-based hybrid-sigma coordinate with levels depth varying on time)

what we did was to compute the height of the vertical layers from different simulation days/hours and then define fixed heights per layer as an average of the results obtained.

The following figures show the average height above ground level (m.a.g.l) of the NMMB-MONARCH model per vertical layer and simulation day/hour (average over the whole domain) and the maximum height difference observed between simulations days/hours in absolute and relative terms. The simulations days that were used were 2015/01/05-00UTC (height_1), 2015/01/05-12UTC (height_2), 2015/07/05-00UTC (height_3) and 2015/07/05-12UTC (height_4).

[Figure]

[Figure]

According to the results, differences between computed heights are not very significant, especially within the boundary layer, where most of the emissions are located. In this sense, we consider that the uncertainty and variability associated with the emission vertical profiles currently available in the literature may be higher. It is also important to highlight that the assumption made in HERMESv3_GR has also been applied in previous works (i.e. assuming fixed vertical layers for emission distribution although the air quality models use sigma levels). The following references are used as example:

Pozzer, A., Jöckel, P., and Van Aardenne, J.: The influence of the vertical distribution of emissions on tropospheric chemistry, Atmos. Chem. Phys., 9, 9417-9432, https://doi.org/10.5194/acp-9-9417-2009, 2009.

Mailler, S., Khvorostyanov, D., and Menut, L.: Impact of the vertical emission profiles on background gas-phase pollution simulated from the EMEP emissions over Europe, Atmos. Chem. Phys., 13, 5987-5998, https://doi.org/10.5194/acp-13-5987-2013, 2013.

Brunner, D., Kuhlmann, G., Marshall, J., Clément, V., Fuhrer, O., Broquet, G., Löscher, A., and Meijer, Y.: Accounting for the vertical distribution of emissions in atmospheric $CO_2$ simulations, Atmos. Chem. Phys. Discuss., https://doi.org/10.5194/acp-2018-956, in review, 2018.

Having said that, we have added the following sentence in order to point out this limitation:

"Note that HERMESv3_GR is currently designed as an off-line model and cannot use or take into account the variability of the vertical layer depth used by atmospheric chemistry models based on sigma vertical coordinates. Consequently, the system cannot distribute the emissions to the exact sigma levels of the models (which slightly vary in time and space) but to a set of fixed vertical levels that are close to them. This assumption is in line with previous modelling works (e.g. Mailler et al., 2013). Moreover, the impact of this limitation can be assumed to be minor when compared to the large uncertainty and variability associated with the emission vertical profiles available in the literature (e.g. Bieser et al., 2011)."

Second, some methods have been developed to temporalize (or even spatialize) the emissions from several sources (for example residential wood burning, agriculture) and such methods are used by some models. I understand it would have been difficult to do in a first approach, but it may be useful to indicate if such methods could be implemented into HERMES.

Response to Reviewer#1 comment No. 2: We understand that the reviewer is referring to methods that use meteorological parameters to derive temporal profiles such as the heating degree day for residential combustion (e.g. Mues et al., 2014) or the parametrizations proposed by Skjøth et al. (2011) for agricultural emissions.

Following with the previous comment, HERMESv3_GR is currently designed as an off-line model and therefore cannot directly take into account the meteorological information provided by atmospheric chemistry models. Nevertheless, meteorological-dependent parametrization to temporally distribute emissions can be indirectly considered within HERMESv3_GR through the application of gridded temporal profiles defined by the user. An example is already provided in Sect. 2.5.3, in which gridded monthly temporal profiles based on meteorological parametrizations and crop calendars are applied to the EDGAR $NH_3$ emissions. Similarly, a user could create a gridded temporal profiles using the heating degree day concept and then apply it to the residential sector emissions.

This concept has been clarified in the text as follows:

"Figure 6 compares the monthly agricultural soil $NH_3$ emissions (March and June 2010) reported by EDGARv432 in East Asia when using its default temporal profile (Figures 6.a and c) and when combined with updated gridded temporal weights that considers the effect of meteorology and crop calendars (Fig. 6b and 6d). These gridded profiles were derived from the monthly inventories reported by Zhang et al. (2018) for China and Paulot et al. (2014) for rest of the world, which seasonality is based on the temporal parametrizations reported by Skjøth et al. (2011)."

"The possibility offered by HERMESv3_GR to use gridded temporal profiles derived from meteorological parametrizations can be extended to other sources such as the residential combustion sector, for which the application of the heating degree day approach has been proved to be effective (e.g. Mues et al., 2014)."

Mues, A., Kuenen, J., Hendriks, C., Manders, A., Segers, A., Scholz, Y., Hueglin, C., Builtjes, P., and Schaap, M.: Sensitivity of air pollution simulations with LOTOS-EUROS to the temporal distribution of anthropogenic emissions, Atmos. Chem. Phys., 14, 939-955, https://doi.org/10.5194/acp-14-939-2014, 2014.

Skjøth, C. A., Geels, C., Berge, H., Gyldenkærne, S., Fagerli, H., Ellermann, T., Frohn, L. M., Christensen, J., Hansen, K. M., Hansen, K., and Hertel, O.: Spatial and temporal variations in ammonia emissions – a freely accessible model code for Europe, Atmos. Chem. Phys., 11, 5221-5236, https://doi.org/10.5194/acp-11-5221-2011, 2011.

The figures should be improved. The scale of the maps should be revised (as the maps are almost entirely blue) to improve the readability and increase the number of details. I would recommend using a log scale to avoid showing only high values, and to not color areas without emissions.

Response to Reviewer#1 comment No. 3: Authors completely agree with the reviewer. We improved all figures using a log scale (to avoid showing only high values) and a "starts-with-white" color bar (to not color areas without emissions). As an illustration, previous and revised versions of Figure 3 are shown below:

Previous version

[Figure]

Current version

[Figure]

In figure 2, the order of the columns (ei, sector, ref_year, active, factor_mask, regrid_mask, pollutants) does not correspond to the order in the text (ei, sector, ref_year, pollutants, active, factor_mask, regrid_mask), making the text a bit difficult to follow.

Response to Reviewer#1 comment No. 3: The order of the columns in Figure 2 has been changed in order to correspond with the order in the text.

Moreover, the examples in figure 2 are difficult to understand without referring to the text.

Response to Reviewer#1 comment No. 4: Title descriptions have been added to the maps in order to facilitate their interpretation. Moreover, the following sentence has been added in the figure caption:

"The corresponding emission inventory configuration files used in each example are shown at the top"

Moreover, I wonder if there could be a mistake in example 1, as the pollutants "nox_no2" and "co" are written in the columns whereas the caption of the figure and the text refer to OC emissions (not CO).

Response to Reviewer#1 comment No. 5: The reviewer is right. The pollutants "nox_no2" and "co" have been replaced by oc.

P8 l26: What does the authors means by first-order conservative? The method of Hill et al. (2004) should be better explained. From what I understand, HERMES does not use the landuse and only distribute homogeneously the emissions and therefore could distribute land emissions over seas or distribute agricultural emissions onto cities.

Response to Reviewer#1 comment No. 6: First-order conservative means that the method preserves the integral of the source field across the regridding. The details of the method have been added in the text as follows:

"The regridding method is first-order conservative, which means that it preserves the integral of the source field across the regridding. The weight for a particular source cell $i$ and destination cell $j$ ($W_{i,j}$) is based on the ratio of the source cell area overlapped with the corresponding destination cell area (Eq. 2):

$$W_{i,j} = f_{i,j} * \frac{AS_i}{AD_j}, \qquad\qquad\qquad\qquad\qquad\qquad\qquad\qquad (4)$$

5   Where $f_{i,j}$ is the fraction of the source cell $i$ contributing to destination cell $j$, and $AS_i$ and $AD_j$ are the areas of the source and destination cells."

As the reviewer points out, no spatial proxies are currently used during the regridding process. The main reason for this is that most of the emission inventories that are currently available in HERMESv3_GR have a spatial resolution that is higher and suitable enough for global and regional air quality modelling (0.1x0.1 degrees or higher in all cases except for ECLIPSEv5

10  and CEDS, which are reported at 0.5x0.5 degrees). As mentioned in the introduction section, these inventories are not meant to be used for urban air quality modelling (i.e. resolutions of 1-5 km$^2$) since they are too coarse (i.e. the spatial proxies used to allocate them are of poor resolution and may not apply to certain emission processes). Having said that, it is true that for some inventories (e.g. ECLIPSEv5, 0.5x0.5 degrees) the application of sector specific spatial proxies during the remapping process could allow improving the emission results.

15  This current limitation of HERMESv3_GR as well as a future task to improve it has been added in the manuscript as follows: "In its current version, HERMESv3_GR does not use any type of spatial proxy (e.g. land use, population data) during the regridding process. The main reason for this is that most of the inventories currently available in the emission data library have a spatial resolution that is higher and suitable enough for global and regional air quality modelling (i.e. 0.1x0.1 degrees). However, for those inventories with low spatial resolution (e.g. ECLIPSEv5a, 0.5x0.5 degree) the application of sector specific

20  spatial proxies may be of importance when performing the remapping onto finer working domains. Future works will focus on improving this limitation by rebalancing the interpolation weights derived from ESMF with spatial proxy-based weight factors."

P8 l31-32: the authors use a gridding country mask to allocate emissions to a specific country. How are separated the emissions when there are several countries into a cell? Some inventories (like the EMEP inventory) directly provide the information of

25  the emitting country. In that case, using a country mask is not useful. Is the information of emitting country use when given?

Response to Reviewer#1 comment No. 7: In its current version, HERMESv3_GR does not consider the information of the emitting country and subsequently emissions are not separated when there are several countries involved into a cell (i.e. border cells). This feature is not included in the tool since most of the original emission inventories considered in HERMESv3_GR do not report this type of information (e.g. EDGAR, HTAP, ECLIPSE and CEDS report total emissions per grid cell but do

30  not specify which fraction corresponds to which country). Hence, we decided to implement a common masking approach that can be applied to any inventory, regardless of the level of information available. This limitation of the tool has been included in the text as follows:

"A current limitation of the masking method is that it does not consider that country border cells may include emissions of more than one country (i.e. it is assumed that all emissions belong to the country that contains the largest fraction of the cell). This limitation is mainly driven by the fact that most of the original inventories do not provide the information of the emitting country (i.e EDGAR, HTAP, ECLIPSE and CEDS report total emissions per grid cell but do not specify which fraction corresponds to which country). Future improvements will include the use of this information when given by the original inventory (i.e. EMEP and TNO_MACC-iii)."

P9 l4-10: several methods are presented to distribute emissions onto vertical layers. A discussion on the comparison of the methods, with the strength and weaknesses of each methods, would be appreciated.

Response to Reviewer#1 comment No. 8: The two methods implemented in HERMESv3_GR to distribute biomass burning emissions across vertical layers are derived from the work by Veira et al., (2015), in which they perform a sensitivity analysis to see the impact of the vertical distribution of forest fire emissions on black carbon concentrations. Although uniform vertical distributions are used in most modelling studies, some works have also showed that fires with high injection heights might emit a large fraction of the emissions into the upper part of the plumes (e.g. Luderer et al., 2006). Given the large uncertainty of this topic, we decided to include both approaches in the model, so that the user can have more flexibility The following information has been added in the manuscript:

"The two approaches are derived from the work by Veira et al., (2015), in which they perform a sensitivity analysis to see the impact of the vertical distribution of forest fire emissions on black carbon concentrations. Although uniform vertical distributions are used in most modelling studies, some works have also showed that fires with high injection heights might emit a large fraction of the emissions into the upper part of the plumes (e.g. Luderer et al., 2006)."

Veira, A., Kloster, S., Schutgens, N. A. J., and Kaiser, J. W.: Fire emission heights in the climate system – Part 2: Impact on transport, black carbon concentrations and radiation, Atmos. Chem. Phys., 15, 7173-7193, https://doi.org/10.5194/acp-15-7173-2015, 2015.

Luderer, G., Trentmann, J., Winterrath, T., Textor, C., Herzog, M., Graf, H. F., and Andreae, M. O.: Modeling of biomass smoke injection into the lower stratosphere by a large forest fire (Part II): sensitivity studies, Atmos. Chem. Phys., 6, 5261-5277, https://doi.org/10.5194/acp-6-5261-2006, 2006.

Speciation mapping: This section lacks details and several elements seem weird. In this state, it gives the impression that the speciation is not treated appropriately. For NMVOCs, I don't understand how it is possible to convert from mass to moles before using the speciation. You would need to know the speciation to compute the mean molar masses of NMCOVs. For NOx, I guess that you use the molar mass of NO2 if the NOx emissions are given as NO2 equivalent.

Response to Reviewer#1 comment No. 9: The speciation process is correctly treated in HERMES. Nevertheless, authors agree with the reviewer that the current section on speciation mapping lacks details. The whole section has been rewritten in order to clarify better how the pollutant-to-species conversion factors have been developed and what is the specific treatment applied to NMVOCs:

"This process converts the pollutants provided in the original emission inventories to the species needed by the atmospheric chemistry model of interest and its corresponding gas phase and aerosol chemical mechanism. The conversion is performed using a speciation CSV file, in which the user defines mapping expressions between the source inventory pollutants and destination chemical species. Each mapping expression defines the pollutant-to-species relationships and factors for converting the input emissions pollutant to the desired model species.

These conversion factors are mass-based (i.e. g of chemical specie · g of source pollutant$^{-1}$) for all source inventory pollutants except for NMVOC, which requires a specific approach (see paragraph below). The factors proposed for $NO_x$ assume a split of 0.9 for NO and 0.1 for $NO_2$ for all sectors (Houyoux et al., 2000) except for road transport and biomass burning, for which specific factors are derived from the works by Burling et al. (2010) and Rappenglueck et al. (2013). In the case of $PM_{2.5}$, the factors are derived from multiple sources of information including the particular matter SPECIATE (Simon et al., 2010) and SPECIEUROPE (Pernigotti et al., 2016) databases and the works by Visschedijk et al. (2007) and Reff et al. (2009). Source specific organic matter (OM) to OC fractions are derived from Klimont et al. (2017). For pollutants that have only one way of being speciated (e.g., mapping the CO pollutant to the CO species) a default factor of 1 is proposed for all sources and inventories. During the chemical speciation process, HERMESv3_GR also performs a conversion from mass to moles for the gas-phase species using a molecular weight CSV file included in the input database of the system. Note that for $NO_x$ two molecular weights are proposed since some inventories report emissions as NO ("nox_no", 30 g·mol$^{-1}$) and some others as $NO_2$ ("nox_no2", 46 g·mol$^{-1}$).

For NMVOC emissions reported as individual chemical compounds (e.g. $C_2H_4O$ in GFASv1.2) or following the GEIA 25 NMVOC groups (e.g. voc15 in EDGARv4.3.2_VOC), the proposed conversion factors are mole-based (i.e. mol of chemical specie · mol of source pollutant$^{-1}$) and are derived from the mechanism-dependent mapping tables developed by Carter (2015). In this case, the conversion from mass to moles of original emissions is performed beforehand, and also using the information of the molecular weight CSV file.

Finally, for NMVOC emissions reported as a single category (i.e. as a sum of $n$ individual chemical compounds) (e.g. EMEP), the conversion factors proposed in HERMESv3_GR for each inventory $i$, pollutant sector $s$ and chemical species $\bar{e}$ ($SF_{\bar{e},s,i}$) were estimated as follows (Eq. 5):

$$SF_{\bar{e},s,i} = \sum_{j=1}^{n} \frac{X_{j,s}}{MW_j} * C_{j,\bar{e}} \, , \tag{5}$$

Where $X_{j,s}$ is the mass fraction of chemical compound $j$ to total NMVOC emissions for source $s$, $MW_j$ is the molecular weight of chemical compound $j$ and $C_{j,i}$ is the mole-based conversion factor of chemical compound $j$ to destination chemical species $\bar{e}$. $X_{j,s}$ values are obtained from the NMVOC SPECIATE database, while $MW_j$ and $C_{j,i}$ where obtained from Carter et al. (2015). The units of resulting proposed conversion factors is mol of chemical specie · g of source pollutant$^{-1}$."

Rappenglueck, B., Lubertino,G., Alvarez, S., Golovko, J., Czader, B., and Ackermann, L.: Radical precursors and related species from traffic as observed and modeled at an urban highway junction, J. Air Waste Manage., 63, 1270–1286, doi:10.1080/10962247.2013.822438, 2013.

Houyoux, M. R., Vukovich, J. M., Coats, C. J., Wheeler, N. J. M., and Kasibhatla, P. S.: Emission inventory development and processing for the Seasonal Model for Regional Air Quality (SMRAQ) project, J. Geophys. Res.-Atmos., 105, 9079–9090, doi: 10.1029/1999JD900975, 2000.

Some explanations on Table2 are needed. I don't understand why: - NO = nox_no2 and NO2=0.18*nox_no - TOL=0.293*voc13 (said to be benzene) +voc14 (said to be toluene) while there is a separate benzene species (and why 0.293).

Response to Reviewer#1 comment No. 10:

NO = nox_no2

5   This was a mistake and has been corrected as follows: NO = 0.84*nox_no2 and NO2 = 0.16* nox_no2

This relationship indicates that 84% of total CEDS $NO_x$ emissions are mapped to the NO RADM2 species and the 16% left to NO2. The weight factors are based on the work by Rappenglueck et al. (2013). The source inventory pollutant is called "nox_no2" because NOx emissions in the CEDS inventory are reported as $NO_2$.

All this information has been added in the paragraph where Table 2 results are discussed.

10   Rappenglueck, B., Lubertino,G., Alvarez, S., Golovko, J., Czader, B., and Ackermann, L.: Radical precursors and related species from traffic as observed and modeled at an urban highway junction, J. Air Waste Manage., 63, 1270–1286, doi:10.1080/10962247.2013.822438, 2013.

NO2=0.18*nox_no

This relationship indicates that 18% of total GFAS $NO_x$ emissions are mapped to the NO2 CB05 species. The original $NO_x$

15   emissions are called "nox_no" because GFAS report them as NO. HERMESv3_GR needs to discriminate between $NO_x$ emissions reported as NO and the ones reported as $NO_2$ since the molecular weight that applies to each case is different.

TOL=0.293*voc13 (said to be benzene) +voc14 (said to be toluene):

The RADM2 chemical mechanism does not have a specific BENZENE species. According to Carter (2015) the benzene chemical compound is mapped to the TOL RADM2 species by multiplying it by 0.293. The following figure, which is a

20   caption of the mechanism-dependent mapping tables developed by Carter (2015) (available at: http://www.engr.ucr.edu/~carter/emitdb/), confirms this fact:

| | A | B | CK | CL | CM | CN | CO | CP | CQ | CR | CS |
|---|---|---|---|---|---|---|---|---|---|---|---|
| 1 | Individual Compounds in DB | | tch there --> | | | | | | | | |
| 2 | 1852 | 11/17/14 | 1.000 | | | 1.215 | | 1.000 | | | 2335 |
| 3 | | | 0.150 | 1803 | 1803 | 0.078 | 1803 | 0.076 | 7 | | |
| 4 | Unique | | Species Assignments for "Lumped Molecule" mechanisms --| | | | | | | | SA |
| 5 | ID | Description | 1 | S11D | RADM2 | | RACM2 | | | | Code |
| 352 | C163702-07-6 | 1,1,1,2,2,3,3,4,4-nonafluoro-4-met | 1.000 | NROG | NROG | 1.000 | NROG | 1.000 | | | 1 |
| 353 | C163702-08-7 | 2-(fl2methoxymethyl)-1,1,1,2,3,3,3 | 1.000 | NROG | NROG | 1.000 | NROG | 1.000 | | | 1 |
| 354 | C138495-42-8 | 1,1,1,2,3,4,4,5,5,5-decafluoropent; | 1.000 | NROG | NROG | 1.000 | NROG | 1.000 | | | 1 |
| 355 | C77-47-4 | hexachlorocyclopentadiene | 1.000 | OTH4 | HC5 | 1.075 | HC5 | 1.000 | | | |
| 356 | C678-26-2 | perfluoropentane | 1.000 | NROG | NROG | 1.000 | NROG | 1.000 | | | 3 |
| 357 | C3638-35-5 | Isopropyl Cyclopropane | 1.000 | OTH3 | HC3 | 0.964 | HC3 | 1.000 | | | 1 |
| 358 | C96-37-7 | methylcyclopentane | 1.000 | MECYCC5 | HC5 | 0.956 | HC5 | 1.000 | | | 1 |
| 359 | C110-82-7 | cyclohexane | 1.000 | CYCC6 | HC8 | 0.945 | HC8 | 1.000 | | | 1 |
| 360 | C110-54-3 | n-hexane | 1.000 | NC6 | HC5 | 0.956 | HC5 | 1.000 | | | 1 |
| 361 | C107-83-5 | 2-methylpentane | 1.000 | M2C5 | HC5 | 0.956 | HC5 | 1.000 | | | 1 |
| 362 | C96-14-0 | 3-methylpentane | 1.000 | M3C5 | HC5 | 0.956 | HC5 | 1.000 | | | 1 |
| 363 | C75-83-2 | 2,2-dimethylbutane | 1.000 | M22C4 | HC3 | 0.964 | HC3 | 1.000 | | | 1 |
| 364 | C79-29-8 | 2,3-dimethylbutane | 1.000 | M23C4 | HC5 | 0.956 | HC5 | 1.000 | | | 1 |
| 365 | C71-43-2 | benzene | 1.000 | BENZ | TOL | 0.293 | BEN | 1.000 | | | 1 |
| 366 | C26519-91-5 | methylcyclopentadiene | 1.000 | OLE2 | OLI | 1.000 | DIEN | 1.000 | | | |
| 367 | C693-02-7 | 1-hexyne | 1.000 | OLE2 | OLI | 1.000 | HC8 | 1.000 | | | |
| 368 | C592-42-7 | 1,5-hexadiene | 1.000 | OLE2 | OLI | 1.000 | DIEN | 1.000 | | | 2 |
| 369 | C42296-74-2 | hexadiene | 1.000 | OLE2 | OLI | 1.000 | DIEN | 1.000 | | | 2 |
| 370 | C7319-00-8 | Trans 1,4-Hexadiene | 1.000 | OLE2 | OLI | 1.000 | DIEN | 1.000 | | | 1 |
| 371 | C592-45-0 | 1,4-hexadiene | 1.000 | OLE2 | OLI | 1.000 | DIEN | 1.000 | | | 2 |
| 372 | C763-30-4 | 2-methyl-1,4-pentadiene | 1.000 | OLE2 | OLI | 1.000 | DIEN | 1.000 | | | 2 |
| 373 | C1120-62-3 | 3-methylcyclopentene | 1.000 | M3CC5E | OLI | 1.000 | OLI | 1.000 | | | 1 |
| 374 | C110-83-8 | cyclohexene | 1.000 | CYCHEXE | OLI | 1.000 | OLI | 1.000 | | | 1 |
| 375 | C693-89-0 | 1-methylcyclopentene | 1.000 | M1CC5E | OLI | 1.000 | OLI | 1.000 | | | 1 |
| 376 | C1759-81-5 | 4-Methylcyclopentene | 1.000 | C6OLE2 | OLI | 1.000 | OLI | 1.000 | | | 2 |
| 377 | C592-48-3 | 1,3-hexadiene | 1.000 | OLE2 | OLI | 1.000 | DIEN | 1.000 | | | 2 |
| 378 | C5194-51-4 | trans,trans-2,4-hexadiene | 1.000 | OLE2 | OLI | 1.000 | OLI | 1.000 | | | 1 |

Master List | Compounds | Notes | Atoms | Mix Profile Assignments | SAROAD assignments

For the first two cases, footnotes have been added to the table. Regarding the last case, the reference to Carter (2015) has been added in the paragraph where the Table 2 is discussed.

A similar question can be asked for almost species. - POA=3*oc (if it is to convert OC emissions into OM emissions, a factor 3 is very high and very unlikely) - EC=5.9*bc (it seems like that the emissions are artificially increase by a factor 5.9) - PMfine = 3.3*pm25-3*oc-5.9*bc (it seems like the mass of PM is artificially increase by a factor 3.3)

Response to Reviewer#1 comment No. 11: This was a mistake and has been corrected as follows:

POA = 1.8*oc (following Klimont et al., 2017)

EC = bc

PMFINE = pm25-oc-bc

Klimont, Z., Kupiainen, K., Heyes, C., Purohit, P., Cofala, J., Rafaj, P., Borken-Kleefeld, J., and Schöpp, W.: Global anthropogenic emissions of particulate matter including black carbon, Atmos. Chem. Phys., 17, 8681-8723, https://doi.org/10.5194/acp-17-8681-2017, 2017.

Writing module: as Figure 7 shows the time for writing increase with the number of processors used. As the authors said, the writing function does not scale properly, probably due to the NetCDF 4 library. Did the authors try to write (if possible) the results with only one processor or the use a specific library (like pnetcdf) for parallel writing?

Response to Reviewer#1 comment No. 12: Following reviewer's recommendation, we adapted HERMESv3_GR so that the

5 writing function can be executed using only one processor (i.e. serial writing). We re-run the scalability test described in the manuscript twice: one executing the writing function in serial and another one in parallel. The results obtained show that for a low number of processors (i.e. from 1 to 48), the parallel writing is faster than the serial one. Nevertheless, for the runs using 96 processors or more, the serial writing becomes faster since its execution time remains almost constant, in contrast to what is experienced with the parallel approach.

10 We have updated the discussion of the results and Figure 7 accordingly:

"The performance of the system when applying the serial writing approach (black line with markers) varies as a function of the processors used. For a low number of cores (i.e. from 1 to 48), the parallel writing is faster than the serial one. Nevertheless, when using 96 processors or more, the serial writing becomes faster since its execution time remains almost constant, in contrast to what is experienced with the parallel approach. This fact allows reducing the total execution time by a factor of up

15 to 1.5 (510 cores). The potential disadvantage of using the serial writing is that for large emission experiments (i.e. large domains) the user may run into memory problems since all the data needs to be treated by a single processor. In the present test, we solved this issue by using all the memory resources of a compute node without sharing them with other users (i.e. 96Gb). Considering the advantages and disadvantages of each method, both the serial and parallel writing approaches are enabled in HERMESv3_GR."

[Figure]

Regarding the use of a specific library for parallel writing (like pnetcdf as suggested by the reviewer), this is a task that we will investigate in the future. In order to make it more clearly, we added the following sentence in the text:

"The low performance of the writing function will be addressed in future versions of HERMESv3_GR. For this, two strategies will be tested, including: (i) the integration of an I/O server that allows writing completed rows in row-major order and (ii) the use of other libraries specific for parallel writing (e.g. pnetcdf)."

Minor comments:

P2 l4: the authors should add a few words on why the global and regional inventories are too imprecise for urban scale modelling

Response to Reviewer#1 comment No. 13: This discussion is already included in the third paragraph on the same page:

"Global and regional inventories are too imprecise for urban scale modelling applications (e.g. Timmermans et al., 2013). Emission and activity factors lack specificity for the local conditions of interest (e.g. Guevara et al., 2014), and the spatial proxies used to allocate the emissions are of poor quality and may not apply to certain emission processes (e.g. Lopez-Aparicio et al., 2017). These inventories are for example limited when it comes to predict and assess the impact of emission reduction measures upon local air quality such as the change of speed limits (e.g. Baldasano et al., 2010) or the penetration of new vehicle technologies (e.g. Soret et al., 2014)."

P9 l11: If you transform a 0.1_x0.1_ inventory into 1_x1.4_ emissions, it is not technically an interpolation. I would not use the word interpolation in the text and only use the word regridding.

Response to Reviewer#1 comment No. 14: Authors agree with the reviewer. The word interpolation has been replaced by regridding in the text.

P9 l22: a.g.l is not defined

Response to Reviewer#1 comment No. 15: The acronym has been defined as above ground level in the revised manuscript.

P12 l5: ":" instead of "Table 2" at the beginning of the line 6

Response to Reviewer#1 comment No. 16: Changed

P15 l30: "which are starting to be widely used in global models"

Response to Reviewer#1 comment No. 17: Changed

Anonymous Referee #2

The paper describes an open source system to process various emission datasets is a flexible manner allowing for changes in projections, scales and making combinations of different inventories. Moreover it provides options for applying different temporal or emission height profiles to generate model-ready emissions input. One of the nice things is that it will allow modelers to relatively easy do sensitivity tests by the ability to scale and/or quickly combine various sets. I do think there is some risk in this, in the sense that people who use it may think that everything is compatible and you can "shop" until you find what you need but in the end this is more a concern than a comment on the paper. The paper is well written and clear. In my opinion it is a good contribution for GMD and I only have minor comments which should be taken into account before accepting the paper

Thank you for the positive and constructive feedback. We completely agree with the comment that users of HERMESv3_GR need to be careful when using and combining emission inventories, and that a clear knowledge of the original inventories is needed. We have addressed this issue in the response to the last comment.

Abstract: please remove "highly" in l 10. It is customizable but highly is an undefined property. What you may find low, someone else may find high and vice versa. This occurs at various places.

Response to Reviewer#2 comment No. 1: Authors agree with the reviewer. The word highly has been removed from the text.

In the introduction P2 L 18 it is stated that "A potential remedy for the latter is to combine different inventories and apply adjustment factors in order to improve the representativeness of the emission data: : :." This should be a bit better explained and possibly also discussed further in the paper. What does improving the representativeness mean? It is important to acknowledge that we should not work towards (and the system is not intended for) having only one totally harmonized inventory. Like models, inventories work from different assumptions with different data and solutions. Having independent datasets is crucial from a science perspective.

Response to Reviewer#2 comment No. 2: Authors completely agree with the reviewer. The sentence was not formulated in a correct way. The objective of HERMESv3_GR is not to improve the representativeness of the inventories, but to give a transparent and flexible framework for their processing when used for air quality modelling. The sentence has been rephrased as follows:

"While having independent emission datasets instead of only one totally harmonized inventory is crucial from a science perspective, having the capacity to combine them and apply adjustment factors in a flexible and transparent way can be also of importance for air quality modelling studies."

P2 l25 I suggest to replace "quality" with resolution – the quality may be good for a global product but not for a regional product.

Response to Reviewer#2 comment No. 3: Authors agree with the reviewer. The word quality has been replaced with resolution

P3 l4 "highly" – see previous comment

Response to Reviewer#2 comment No. 4: Removed

P6 l 6-7 does the user provide data? Or the data provider? I assume there can be users who do not provide data?

Response to Reviewer#2 comment No. 5: All the pre-processing functions used to transform the original inventories are included in the HERMESv3_GR repository. Nevertheless, the original emission inventories are not stored inside the HERMESv3_GR database and users need to download them from the corresponding data provider (e.g. EDGAR emission inventories need to be downloaded from http://edgar.jrc.ec.europa.eu/).

We have decided to proceed this way for two main reasons: i) some of the emission inventories that HERMESv3_GR can process cannot be passed on to third parties without the data provider's consent and ii) we believe it is good practice that users access the original files through the official source of information, so that the data providers can monitor the usage of their datasets. With the aim of helping the users, the HERMESv3_GR wiki contains a section that provides information of each emission inventory, including reference and downloading website/contact person (https://earth.bsc.es/gitlab/es/hermesv3_gr/wikis/user_guide/emission_inventories). This information is also included in a README section inside each pre-processing function (e.g. https://earth.bsc.es/gitlab/es/hermesv3_gr/blob/production/preproc/edgarv432_ap_preproc.py)

We believe this was not explained clearly enough and subsequently we have added the following paragraph in the revised version of the manuscript:

"It is important to note that the original gridded emission inventories are not stored inside the HERMESv3_GR database and that users need to download them from the corresponding data provider's platform (e.g. EDGAR inventories are obtained from http://edgar.jrc.ec.europa.eu/). This decision is based on the fact that: i) some of the emission inventories that HERMESv3_GR can process cannot be passed on to third parties without the data provider's consent and ii) we believe it is good practice that users access the original files through the official source of information, so that the data providers can monitor the usage of their datasets. With the aim of helping the users, the HERMESv3_GR wiki contains a section that provides information of each emission inventory, including the official downloading website/contact person (see Sect. **5**). This information is also included in a README section inside each pre-processing function."

P8 l 1-3 – This possible explanation should be removed. As it is not further documented it remains speculation and does not belong in this paper. Furthermore, for making comparisons between a certain emission category from different inventories one should not use maps but the emission data by sector.

Response to Reviewer#2 comment No. 6: Authors agree with the reviewer. The explanation has been removed.

P 12 l 6 – reference to Table 2 is missing at the start of the sentence.

Response to Reviewer#2 comment No. 7: Reference to Table 2 has been added.

P12 l 14-15 – please check if sentence is correct it sort of says that NO is mapped to NO2 but maybe I misunderstand.

Response to Reviewer#2 comment No. 8: The sentence was wrong. It has been corrected as follows:

"$NO_x$ emissions (which are originally reported as $NO_2$) are mapped to NO and NO2 using mass-based conversion factors of 0.84 ("nox_no2*0.84") and 0.16 ("nox_no2*0.16") (Rappenglueck et al., 2013)"

Rappenglueck, B., Lubertino,G., Alvarez, S., Golovko, J., Czader, B., and Ackermann, L.: Radical precursors and related species from traffic as observed and modeled at an urban highway junction, J. Air Waste Manage., 63, 1270–1286, doi:10.1080/10962247.2013.822438, 2013.

Table 2 has also been corrected according to the new text.

5    P15 l 12 "and temperature" is not correct maybe you mean "driven by temperature". The sentence now implies that temperature is a pollutant sector. Also pollutant sector should be source sector.

Response to Reviewer#2 comment No. 9: Authors agree with the reviewer. The sentence has been changed following the reviewer's suggestions.

P15 l22 remove "–"

10    Response to Reviewer#2 comment No. 10: Removed

P15 l 25 work not works

Response to Reviewer#2 comment No. 11: Changed

P15 l 30 widely USED in

Response to Reviewer#2 comment No. 12: Changed

15    Figures: At least when printed the maps are not very clear and while they only serve as an illustration it seems the legend is not well chosen. It would be better to show more gradients.

Response to Reviewer#2 comment No. 13: Authors completely agree with the reviewer. We improved all figures using a log scale (to avoid showing only high values) and a "starts-with-white" color bar (to not color areas without emissions). As an illustration, previous and revised versions of Figure 3 are shown below:

20    Previous version

[Figure]

Current version

[Figure]

Finally in the conclusions it should be considered to make disclaimer or statement that the system PROCESSES emissions

5    data, it does not make them better. Users should always remain aware that combining parts from different inventories can also

lead to substantial errors because the definition what is included or excluded in certain sectors and/or inventories can differ substantially. A notorious example is e.g. agricultural waste burning which is sometimes included under agriculture sometimes excluded (and than given under waste, or not at all as it is assumed it comes from the Fire emission inventories). So combining apples and oranges without going to the original descriptions of what is included should be avoided. In the end this is the

5 responsibility of the user but a word of warning is warranted.

Response to Reviewer#2 comment No. 14: Authors completely agree with the reviewer. The following statement has been added to the conclusions section:

[revised manuscript text omitted]

 • YYYYMMDD: 20150101
 • YYYYMMDDhh: 2015010100
 • YYYYYMMDD.hh: 20150101.00
 • YYYY/MM/DD: 2015/01/01
 • YYYY/MM/DD_hh: 2015/01/01_00
 • YYYY/MM/DD_hh:mm:ss: 2015/01/01_00:00:00
 • YYYY/MM/DD hh:mm:ss: 2015/01/01 00:00:00
 • YYYY-MM-DD_hh: 2015-01-01_00
 • YYYY-MM-DD_hh:mm:ss: 2015-01-01_00:00:00
 • YYYY-MM-DD hh:mm:ss: 2015-01-01 00:00:00 |
| end_date = 2010/01/02 00:00:00 | [OPTIONAL]Ending date of the simulation (in UTC). If it is not set then end_date = start_date. |
| output_timestep_type = hourly | Temporal resolution of the output file. The options are:
 • Hourly
 • Daily
 • Monthly |
| output_timestep_num = 24 | Number of time steps to simulate |
| output_timestep_freq = 1 | Frequency between time steps |
| [DOMAIN] | |
| output_model = CMAQ | Defines the format of the output emission file as a function of the atmospheric chemistry model conventions. Current options are:
 • MONARCH
 • CMAQ
 • WRF_CHEM |

| | |
|---|---|
| output_attributes = /data/cmaq_global_attributes.csv | Path to the file that contains the global attributes that need to be included in the output NetCDF file according to the corresponding chemical transport model |
| domain_type= lcc | Defines the grid projection on which the emission fields will be generated. Options are:
• global: regular lat-lon grid
• rotated: rotated lat-lon grid
• lcc: lambert conformal conic grid
• mercator: mercator grid |
| vertical_description = /data/profiles/vertical/ vert.csv | Path to the file that contains the vertical description of the desired output |
| aux_files_path = /data/aux_files/<domain_type>_<res> | Path to the directory where the necessary auxiliary files (e.g. timezones file) will be created if they do not exist. If they already exist, HERMESv3_GR will just read them |
| # if domain_type == global:
    inc_lat = 0.5
    inc_lon = 0.703125 | Parameters that define a global regular lat-lon grid:
• inc_lat: Latitudinal grid resolution (degrees)
• inc_lon: Longitudinal grid resolution (degrees). |
| # if domain_type == rotated:
    centre_lat = 35
    centre_lon = 20
    west_boundary = -51
    south_boundary = -35
    inc_rlat = 0.1
    inc_rlon = 0.1 | Parameters that define a regional rotated lat-lon grid:
• centre_lat = Central geographic latitude of the grid (non-rotated degrees).
• centre_lon = Central geographic longitude of grid (non-rotated degrees, positive east).
• west_boundary = Grid's western boundary from center point (rotated degrees).
• south_boundary = Grid's southern boundary from center point (rotated degrees).
• inc_rlat = Latitudinal grid resolution (rotated degrees).
• inc_rlon = Longitudinal grid resolution (rotated degrees). |
| # if domain_type == lcc:
    lat_1 = 37
    lat_2 = 43
    lon_0 = -3
    lat_0 = 40
    nx = 278
    ny = 298
    inc_x = 1000
    inc_y = 1000
    x_0 = 253151.59375
    y_0 = 43862.90625 | Parameters that define a regional lambert conformal conic grid:
• lat_1 = Standard parallel 1 (in degrees).
• lat_2 = Standard parallel 2 (in degrees).
• lon_0 = Longitude of the central meridian (in degrees).
• lat_0 = Latitude of the origin of the projection (in degrees).
• nx = Number of grid columns.
• ny = Number of grid rows.
• inc_x = X-coordinate cell dimension (in meters).
• inc_y = Y-coordinate cell dimension (in meters).
• x_0 = X-coordinate origin of grid (in meters).
• y_0 = Y-coordinate origin of grid (in meters). |

| | |
|---|---|
| # if domain_type == mercator:
   lat_ts = -2.84
   lon_0 = -79.16
   nx = 99
   ny = 81
   inc_x = 1000
   inc_y = 1000
   x_0 = -49500.13899
   y_0 = -355986.692 | Parameters that define a regional lamber conformal conic grid:
• lat_ts = Latitude of true scale (in degrees).
• lon_0 = Longitude of projection center (in degrees).
• nx = Number of grid columns.
• ny = Number of grid rows.

[revised manuscript text omitted]

**Figure 7: Schematic representation of the parallelization of the emission core module of HERMESv3_GR (top) and computational times obtained for each functionality (regrid, vertical, temporal, speciation and writing-parallel/writing-serial) for the scalability test performed (bottom). The destination working domain is divided into vertical sections, according to the number of processors to be used (P_0, P_1, …). Vertical (v0, v1, …) and temporal (t0, t1, …) weight factors are applied to each section in order to transform the 2D arrays (longitude, latitude) into 4D arrays (time, vertical layer, longitude, latitude).**